# Intersectin-1 enhances calcium-dependent replenishment of the readily releasable pool of synaptic vesicles during development

Yi-Mei Yang[1,2,3] , Adam Fekete[1,2], Jason Arsenault[1,2] , Ameet S. Sengar[1], Jamila Aitoubah[1,2], Giovanbattista Grande[1,2], Angela Li[1], Eric W. Salter[1,2,4], Alex Wang[1,5] , Melanie D. Mark[6], Stefan Herlitze[6], Sean E. Egan[7,8], Michael W. Salter[1,2] and Lu-Yang Wang[1,2]

[1]*Neurosciences and Mental Health, SickKids Research Institute, Toronto, Ontario, Canada*

[2]*Department of Physiology, University of Toronto, Toronto, Ontario, Canada*

[3]*Department of Biomedical Sciences, University of Minnesota, Duluth, Minnesota, USA*

[4]*Department of Neuroscience, Brown University, Providence, Rhode Island, USA*

[5]*Department of Neuroscience, Yale University, New Haven, Connecticut, USA*

[6]*Department of Zoology and Neurobiology, Ruhr-University Bochum, Bochum, Germany*

[7]*Cell Biology, SickKids Research Institute, Toronto, Ontario, Canada*

[8]*Department of Molecular Genetics, University of Toronto, Toronto, Ontario, Canada*

Handling Editors: Katalin Toth & Samuel Young

The peer review history is available in the Supporting Information section of this article (https://doi.org/10.1113/JP286462s#support-information-section).

**Abstract figure legend** During development, Intersectin-1 (Itsn1) translocates near voltage-gated calcium channels (VGCCs) to enhance $Ca^{2+}$-dependent replenishment of readily releasable synaptic vesicles (SVs). In the presynaptic terminal, SVs undergo a cycle involving formation (step 1), docking (step 2), exocytosis (step 3), endocytosis (step 4) and refilling (step 5) to sustain transmitter release. The developmental repositioning of Itsn1 (purple) to the vicinity of VGCCs, forming a $Ca^{2+}$ domain (pink), potentially facilitates recovery from exocytosis and thereby supports faithful synaptic transmission during repetitive activity.

Y. Yang and A. Fekete contributed equally to this study.

**The Journal of Physiology**

**Abstract** Intersectin-1 (Itsn1) is a scaffold protein that plays a key role in coupling exocytosis and endocytosis of synaptic vesicles (SVs). However, it is unclear whether and how Itsn1 regulates these processes to support efficient neurotransmission during development. To address this, we examined the calyx of Held synapse in the auditory brainstem of wild-type and Itsn1 mutant mice before (immature) and after (mature) the onset of hearing. Itsn1 was present in the pre- and postsynaptic compartments at both developmental stages. Loss of function of Itsn1 did not alter presynaptic action potentials, $Ca^{2+}$ entry via voltage-gated $Ca^{2+}$ channels (VGCCs), transmitter release or short-term depression (STD) induced by depletion of SVs in the readily releasable pool (RRP) in either age group. Yet, fast $Ca^{2+}$-dependent recovery from STD was attenuated in mature mutant synapses, while it was unchanged in immature mutant synapses. This deficit at mature synapses was rescued by introducing the DH–PH domains of Itsn1 into the presynaptic terminals. Inhibition of dynamin, which interacts with Itsn1 during endocytosis, had no effect on STD recovery. Interestingly, we found a developmental enrichment of Itsn1 near VGCCs, which may underlie the Itsn1-mediated fast replenishment of the RRP. Consequently, the absence of Itsn1 in mature synapses led to a higher failure rate of postsynaptic spiking during high-frequency synaptic transmission. Taken together, our findings suggest that Itsn1 translocation to the vicinity of VGCCs during development is crucial for accelerating $Ca^{2+}$-dependent RRP replenishment and sustaining high-fidelity neurotransmission.

(Received 3 May 2024; accepted after revision 6 September 2024; first published online 10 October 2024)
**Corresponding authors** Y. M. Yang, M. W. Salter and L. Y. Wang: Neurosciences and Mental Health, SickKids Research Institute, Toronto, Ontario, Canada. Email: ymyang@umn.edu, mike.salter@utoronto.ca, luyang.wang@utoronto.ca

**Key points**

- Itsn1 is expressed in the pre- and postsynaptic compartments of the calyx of Held synapse.
- Developmental upregulation of vesicular glutamate transporter-1 is Itsn1 dependent.
- Itsn1 does not affect basal synaptic transmission at different developmental stages.
- Itsn1 is required for $Ca^{2+}$-dependent recovery from short-term depression in mature synapses.
- Itsn1 mediates the recovery through its DH–PH domains, independent of its interactive partner dynamin.
- Itsn1 translocates to the vicinity of presynaptic $Ca^{2+}$ channels during development.
- Itsn1 supports high-fidelity neurotransmission by enabling rapid recovery from vesicular depletion during repetitive activity.

# Introduction

During early postnatal development, the sensory system evolves rapidly, enabling an individual to interact with the environment. The critical period for sensory development is marked by activity-dependent remodelling of synaptic structure and function for enhancing the brain's capacity to process a large amount of information. This process is exemplified in the central auditory system where subtle interaural level differences (ILDs) and interaural timing differences (ITDs) are accurately computed for sound localization (Borst & Soria van Hoeve, 2012; Grothe et al., 2010; Joris & Trussell, 2018). The calyx of Held synapse in the brainstem is an important nucleus in the sound

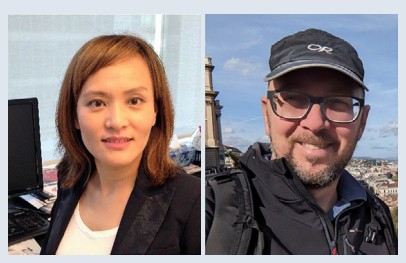

**Yi-Mei Yang** received a PhD from Huazhong University of Science and Technology and postdoctoral training at the University of Toronto. She is currently an Associate Professor at the University of Minnesota, studying the developmental plasticity of synapses. **Adam Fekete**, with a medical background and a PhD from Semmelweis University in Budapest, has expertise in cellular and molecular neuroscience. As a postdoctoral fellow at the University of Victoria, he specialized in synaptic physiology. Currently, he is a Research Associate at the SickKids Research Institute. Using electrophysiology and advanced fluorescence microscopy, he studies the nanoscale mechanisms underlying synaptic transmission and plasticity.

localization pathway. It converts excitatory inputs from ipsilateral globular bushy cells into inhibitory outputs to the contralateral superior olive, where ILDs and ITDs are encoded and sent to other auditory centres (Borst & Soria van Hoeve, 2012; Schneggenburger & Forsythe, 2006).

In mice, the sensitive period for auditory development is defined by the opening of ear canals at postnatal day (P) 12. An exposure to external sound drives profound changes in the calyx of Held synapse over a few days (Wang et al., 2009). Morphologically, the calyx nerve terminals are spoon-shaped before hearing onset but become fenestrated with fine branches after that (Borst & Soria van Hoeve, 2012; Fekete et al., 2019; Grande & Wang, 2011; von Gersdorff & Borst, 2002). Functionally, these synapses develop into a reliable relay station to convey auditory information with high speed and high fidelity, as required for sound localization. To meet such a demand, both pre- and postsynaptic adaptations take place. These include narrowing of action potentials (APs) to reduce the refractory period (Taschenberger & von Gersdorff, 2000; Yang & Wang, 2006), alteration in the subtype, density and location of voltage gated $Ca^{2+}$ channels (VGCCs) (Iwasaki & Takahashi, 1998; Young & Veeraraghavan, 2021), and shortening of their spatial distance to synaptic vesicles (SVs) to enhance release efficiency (Fedchyshyn & Wang, 2005; Yang et al., 2010). Meanwhile, postsynaptic changes, such as the downregulation of NMDA receptors and switching of AMPA receptors from slow-gating GluA1 to fast-gating GluA4 subunits, significantly accelerate excitation–spike coupling (Joshi et al., 2004, 2007; Koike-Tani et al., 2005; Yang et al., 2011). During repetitive activity, mature calyx of Held synapses exhibit less short-term depression (STD) and faster recovery from STD than immature ones (Joshi & Wang, 2002; Taschenberger & von Gersdorff, 2000). As STD is largely accounted for by the depletion of SVs in the readily releasable pool (RRP) (von Gersdorff et al., 1997), the recovery from STD is used to measure the kinetics of RRP replenishment. It has been shown that high-frequency activity gives rise to fast $Ca^{2+}$-dependent replenishment of the RRP (Wang & Kaczmarek, 1998). Although fast recovery from STD is essential for sustaining transmitter release, our understanding of the molecular mechanisms underlying its developmental emergence is limited.

Intersectins (Itsn) are a family of evolutionarily conserved proteins, encoded by *Itsn1* and *Itsn2* genes, each of which has a short and a long isoform (Gubar et al., 2013; Herrero-Garcia & O'Bryan, 2017). Both isoforms contain EH and SH3 domains that bind to synaptic proteins including SNAP25 and dynamin. Itsn1 promotes clathrin-mediated endocytosis by interacting with dynamin (Sengar et al., 1999; Wang et al., 2008; Winther et al., 2013). The long isoform has three additional domains in its C-terminus: Dbl homology

(DH), pleckstrin homology (PH) and C2. The DH–PH domain cassette stimulates activation of Cdc42, a small GTPase of the Rho family involved in actin polymerization during exocytosis (Hussain et al., 2001). The C2 domain may bind $Ca^{2+}$ and play a role in both exocytosis and endocytosis (Gundelfinger et al., 2003; Zhang et al., 2013). Notably, the Itsn1 long isoform is enriched in neurons (Hussain et al., 1999). Itsn1 over-expression in humans is associated with Down syndrome (Pucharcos et al., 1999), while Itsn1 mutation in mice produces a learning and memory deficit, probably due to a loss of interhemispheric connections (Sengar et al., 2013). Using Itsn1 mutants, previous studies have identified the role of Itsn1 in coupling the exocytosis of SVs with their endocytosis (Gerth et al., 2017; Gubar et al., 2013; Japel et al., 2020). For example, Sakaba et al. (2013) showed that deletion of Itsn1 impaired SV replenishment when a constant depolarizing step was applied to evoke $Ca^{2+}$ current and deplete fast-releasing SVs at the immature calyx of Held synapses. However, it remains unknown whether the synaptic function of Itsn1 is developmentally regulated, and if so, how it impacts the excitation–spike coupling in response to physiological stimuli.

To this end, we examined the auditory synapse in wild-type (WT) and Itsn1 mutant (Mut) mice at different developmental stages, P7–11 (before hearing onset, termed 'immature') and P16–20 (after hearing onset, termed 'mature'). By generating natural presynaptic APs to deplete the RRP, we found that Itsn1 was particularly important for $Ca^{2+}$-dependent replenishment of the RRP in mature synapses, but not in immature ones. Imaging analysis further revealed that during development, Itsn1 moved closer to VGCCs, enabling faithful neurotransmission by supporting rapid RRP refilling after intense neural activity.

## Methods

### Ethical approval

Mice were housed in a facility accredited by the Canadian Council on Animal Care. All procedures were approved by the SickKids Animal Care Committee (protocol #1000065180).

### Animals

Mice were kept under a 12 h light–dark cycle (light on from 07.00 to 19.00 h) and reared 2−3 per cage with food and water *ad libitum*. Itsn1 mutants were generated via gene-trap mutagenesis (Sengar et al., 2013). Embryonic stem cells obtained from the German Gene Trap Consortium (GGTC) were implanted into recipient surrogate mice and subsequently backcrossed with 129/Sv

mice for 10 generations. This strategy resulted in a complete loss of function of the *Itsn1* gene, confirmed by western blot analysis of whole brain lysates (Sengar et al., 2013). Genotyping to identify alleles was performed using the primers: ATCACACTCAGTCTTCGCTAGCTG, CCCTACTTGCCTTGGTCTTTGCTT and CGCCTTAT CCGGTAACTATCGTCT. Age-matched WT littermates were used as controls. A knock-in mouse line, in which P/Q-type $Ca^{2+}$ channels were tagged with citrine (*Cacna1a^Citrine*), was provided by Drs Melanie Mark and Stefan Herlitze (Mark et al., 2011).

## Euthanasia

Decapitation was used to collect brain tissue. The mouse was restrained in a plastic cone and quickly decapitated with a sharp single-use blade (009 RD; VWR, Mississauga, Canada). This method ensured rapid loss of consciousness and protected the tissue from chemical contamination.

## Slice preparation

Brain slices from Itsn1 mutant and WT mice of either sex at postnatal days 7−20 were prepared. Following decapitation, their brains were dissected in semi-frozen artificial cerebrospinal fluid (ACSF) including (in mM): NaCl (125), KCl (2.5), glucose (10), $NaH_2PO_4$ (1.25), sodium pyruvate (2), *myo*-inositol (3), ascorbic acid (0.5), $NaHCO_3$ (26), $MgCl_2$ (1) and $CaCl_2$ (2), balanced to a pH of 7.3 when oxygenated with 95% $O_2$ and 5% $CO_2$. Transverse sections of the auditory brainstem containing the medial nucleus of the trapezoid body (MNTB) were sliced to a thickness of 150−250 μm using a Leica VT1200 S vibratome. The slices were then incubated at 35°C for 1 h and maintained at room temperature (∼23°C) before experiments (Fekete et al., 2019; Yang et al., 2014).

## Electrophysiology

Recordings were acquired at a filtering frequency of 4 kHz using a MultiClamp 700B dual-channel amplifier (Molecular Devices, Sunnyvale, CA, USA) and digitized at a sampling rate of 50 kHz with a Digidata 1550B (Molecular Devices). The ACSF used for recordings was supplemented with bicuculline (10 μM) and strychnine (1 μM) to block inhibitory inputs. Excitatory postsynaptic currents (EPSCs) were evoked using a bipolar platinum electrode to stimulate afferent axons. Stimulation paradigms were delivered via a Master-8 stimulator (A.M.P.I., Jerusalem, Israel), with stimulation voltage adjusted to 30−50% above the threshold (typically 3−10 V). The intracellular solution for EPSC recording included (in mM): potassium

gluconate (97.5), CsCl (32.5), EGTA (5), Hepes (10), $MgCl_2$ (1), tetraethylammonium (TEA; 30) and lidocaine N-ethyl bromide (3), adjusted to pH 7.2 with KOH. For recording APs in current clamp mode, the intracellular solution contained (in mM): potassiuim gluconate (97.5), KCl (32.5), EGTA (0.5), Hepes (40), $MgCl_2$ (1), ATP-Na (2) and GTP-Na (0.5), adjusted to pH 7.3 with KOH. The same solution was used to deliver EGTA (increased to 10 mM) and DH−PH fragment (5 μM) into the calyces. For recording $Ca^{2+}$ current, tetrodotoxin (0.5 μm), TEA (10 mM) and 4-aminopyridine (0.3 mM) were added to the ACSF to block $Na^+$ and $K^+$ channels. The intracellular solution for $Ca^{2+}$ current recording included (in mM): CsCl (110), EGTA (0.5), Hepes (40), $MgCl_2$ (1), ATP-Na (2), GTP-Na (0.5), phosphocreatine (12), TEA (20) and potassium glutamate (3), adjusted to pH 7.3 with CsOH. $Ca^{2+}$ current was elicited by a voltage command as depicted in the figure legend. An online P/4 protocol was used for leak subtraction. The holding potential was set at −80 mV for presynaptic terminals and −60 or −70 mV for postsynaptic neurons. Patch electrodes had resistances of 4−6 MΩ for presynaptic and 2.5–3 MΩ for postsynaptic recordings, with series resistances compensated to 90% and kept below 10 and 5 MΩ, respectively. Recordings not reaching an initial GΩ seal or having a holding current over 300 pA were excluded. Rigorous criteria, as described previously (Yang & Wang, 2006; Yang et al., 2014), were applied to ensure recording quality. Reagents were sourced from Sigma (St Louis, MO, USA), Tocris (Ellisville, MO, USA) and Alomone Labs (Jerusalem, Israel).

The course of EPSC decay time, STD and steady-state potentials evoked by current steps was fitted with a single exponential function $f(t) = Ae^{-t/\tau} + C$. The number of spikes evoked by current steps was fitted with a charge–voltage Boltzmann function $f(I) = V_{max}/(1 + e^{(Imid-I)/Ic}) + C$.

The RRPs at immature and mature synapses were depleted using 100 and 300 Hz stimulus trains, respectively. The size of the RRP was quantified using a method detailed previously (Wesseling & Lo, 2002). This method accounts for the rate at which SVs are recruited to the RRP during steady-state release, observed in the later part of stimulus trains. To calculate the RRP size and dynamics of SV recruitment, the following equations were used:

$$fe = \frac{r(1)}{r(\infty)} * \left(1 - e^{\frac{-a}{v}}\right)$$

$$fe = \frac{r(1)}{\sum_{i=1}^{s} r(i) * e^{\frac{-a(s-i)}{v}}}$$

where *fe* is the fusion efficiency, $r(1)$ is the amplitude of the first response, $r(\infty)$ is the amplitude of the steady-state

response, $\alpha$ is the filling rate, $s$ is the number of stimuli and $v$ is the stimulation frequency. The number of SVs in the RRP ($N$) was derived from the following equation:

$$N = \sum_{i=1}^{s} r(i) - w(S)$$

where $w(S)$ is the number of SVs released from the reserve pool during the train.

To assess the RRP replenishment rate, two separate stimulus trains were delivered at varying intervals, with the first train depleting the RRP and the second train measuring recovery from depletion. The amplitude of each EPSC was first subtracted by the average amplitude of the final four (for 100 Hz train) or five (for 300 Hz train) EPSCs in the second train. This subtraction aimed to minimize the impact of ongoing replenishment during steady-state release. The total amplitude of EPSCs triggered by the second train was then divided by that of EPSCs from the first train, yielding a recovery percentage. Lastly, the recovery time course was modelled using an exponential cumulative distribution function comprising one or two components, with each component following the equation $f(t) = p(1 - e^{-t/\tau}) + C$. All functions and parameters are provided in the figures.

### Protein production

Itsn1 DH–PH fragment was generated by expressing glutathione *S*-transferase (GST) fused to Itsn1 amino acids 1216–1592 (Accession #AAD19749.1) using the pGEX-2TK plasmid and BL21 bacterial cells, as per the manufacturer's instructions (Millipore Sigma). The bacterial cell lysate was passed twice through a glutathione sepharose column, followed by two washes. The fragment was then cleaved from the GST bound to the column using thrombin (Millipore Sigma) and eluted according to the manufacturer's protocol. The concentration was measured using the bicinchoninic acid (BCA) assay (Thermo Fisher Scientific, Waltham, MA, USA). The fragment sequence is: SDLHLLDMLTPTERKRQGYIHELIVTEENYVNDLQLV TEIFQKPLTESELLTEKEVAMIFVNWKELIMCNIKLLK ALRVRKKMSGEKMPVKMIGDILSAQLPHMQPYIRFC SCQLNGAALIQQKTDEAPDFKEFVKRLAMDPRCKG MPLSSFILKPMQRVTRYPLIIKNILENTPENHPDHSHL KHALEKAEELCSQVNEGVREKENSDRLEWIQAHVQ CEGLSEQLVFNSVTNCLGPRKFLHSGKLYKAKSNKEL YGFLFNDFLLLTQITKPLGSSGTDKVFSPKSNLQYKM YKTPIFLNEVLVKLPTDPSGDEPIFHISHIDRVYTLRA ESINERTAWVQKIKAASELYIETEKKKREKAYLVRSQR ATGIGRLM.

### Western blot

Quantitative western blotting was performed as previously described (Arsenault et al., 2016). Mouse brainstems were sectioned (500 μm) and MNTB regions were cut out with fine needles under a microscope (Nikon SMZ-2T) from WT and Itsn1 mutant mice. Isolated MNTB regions were homogenized in an ice-cold solution containing 50 mM Tris-HCl, 1% SDS, pH 7.4, supplemented with protease inhibitor cocktail (Roche, Indianapolis, IN, USA) using polypropylene RNase-Free Disposable Pellet Pestles (12-141-368; Fisherbrand). The protein concentration was determined using the BCA assay (Sigma). Equal amounts of protein (3.2 μg) were loaded onto a 10% polyacrylamide-SDS gel and transferred onto a nitrocellulose membrane after electrophoresis. The membranes were blocked in 5% milk for 1 h and incubated at 4°C overnight with primary antibodies: mouse anti-Itsn1 (1:1000; 611574; BD Transduction Laboratories), mouse anti-dynamin (1:1000; 610245; BD Transduction Laboratories) and mouse anti-GAPDH (1:40,000; G8795; Sigma). After rinsing, a goat anti-mouse HRP-conjugated secondary antibody (1:4000; Jackson ImmunoResearch, West Grove, PA, USA) was applied for 2 h. The immunoreactive proteins were visualized using the FluorChem MultiImage Light Cabinet (Alpha Innotech, San Leandro, CA, USA). Densitometric analysis was carried out using the AlphaEaseFC software (Alpha Innotech). The band intensity was normalized to GAPDH and expressed as a percentage of WT P18 expression levels.

### Anterograde tracing

We pulled pipettes with a resistance of 7−8 MΩ from borosilicate glass capillaries (World Precision Instruments, Sarasota, FL, USA) and coated their tips with dextran-conjugated Alexa Fluor 594 (A594d, 10,000 MW, dissolved in 1.5% bovine serum albumin). Coated tips were then inserted into the midline of brainstem sections prepared from *Cacna1a*$^{Citrine}$ mice (Delaney, 2010; Fekete et al., 2019; Mark et al., 2011). We held the tips in place until the dye crystals fully dissolved. Brainstem sections, 250 μm thick, were prepared as described earlier in electrophysiology studies. Slices were incubated for 30 min at 35°C and then 60 min at 23°C, allowing for labelling of axons and terminals within the calyx of Held.

### Immunohistochemistry

Slices were fixed in 4% paraformaldehyde in phosphate-buffered saline (PBS) for 1 h at 24°C and rinsed three times with PBS. After permeabilization with 0.2% Triton X-100 and blocking using a mouse on mouse detection kit (M.O.M kit, Vector Laboratories,

Burlingame, CA, USA), slices were incubated overnight at 4°C with a primary antibody against Itsn1 (mouse anti-Intersectin/ESE-1; 1:200; AB_399020; BD Biosciences), followed by several rinses and a 2 h incubation with a secondary antibody conjugated to Alexa Fluor 647 (A647, rabbit anti-mouse IgG; 1:200; AB_2535808; Thermo Fisher Scientific). For co-immunolabeling experiments, the same procedure was repeated for Itsn1 but using a 1:500 dilution of secondary antibody (A647). After several rinses, slices were blocked with normal goat serum and incubated overnight at 4°C with a primary antibody against vesicular glutamate transporter-1 (guinea pig anti-vGlut1; 1:1000; AB_2301751; Millipore Sigma), followed by several rinses and a 2 h incubation with a secondary antibody conjugated to Alexa Fluor 488 (A488, goat anti-guinea pig IgG; 1:500; AB_2534117; Thermo Fisher Scientific). Slices were mounted on microscope slides and sealed with Prolong Diamond antifade mountant (Thermo Fisher Scientific) and coverslips (1.5H, Marienfeld Superior). To minimize variability in staining, slices from different genotypes and age groups were synchronously labelled using the same master solutions.

## Confocal imaging

To assess the relative abundance of Itsn1 between different genotypes and ages, we used a Carl Zeiss LSM 710 NLO multiphoton laser scanning microscope, equipped with 488 nm argon- and 633 nm helium–neon lasers, to obtain $z$-stack images (0.5 μm per step) of brainstem slices labelled with vGlut1 and Itsn1 (A488 and A647, respectively). Slices were scanned through a 63× oil-immersion objective (1.4 NA, $x$–$y$ pixel spacing = 65.9 nm) and an MBS 488/543/633 dichroic mirror. The emission bandwidth for each fluorochrome was set using a prism-based spectral detector (495–564 nm for A488; 639−723 nm for A647). The system was operated via Zen 2009 software. To minimize variability, we imaged slices from both genotypes and age groups (WT /P8-10 and /P16, Itsn1 mutant /P8-10 and /P16) on the same day using the same settings.

To quantify the distribution of Itsn1 relative to VGCC clusters from *Cacna1a*$^{Citrine}$ mice, we acquired $z$-stack images (0.3 μm per step) using a Leica TCS SP8 confocal imaging system, with a Leica DMI6000 inverted microscope and super-sensitive hybrid detectors. Laser beams of 515, 598 and 648 nm generated from a white light laser were sequentially scanned across the sample through a 63× oil-immersion objective (1.4 NA, $x$–$y$ pixel spacing = 64.6 ± 1.1 nm, $n$ = 40) to excite citrine, A594 and A647 indicators, respectively. The emission bandwidth for each fluorochrome was set using a prism-based spectral detector (520−570 nm, citrine; 603−635 nm, A594; 658−749 nm, A647). The dye and autofluorescence were separated using the fluorescence lifetime gating function (0.3−6.0 ns). The system was operated via Leica LAS X software. To minimize variability, we imaged slices from both age groups (*Cacna1a*$^{Citrine}$ /P9-10 and /P16-17) on the same day using the same settings.

To analyse imaging data, we used Zen 2009 (Carl Zeiss, Oberkochen, Germany), Microsoft Excel, Igor Pro NeuroMatic package and ImageJ (http://imagej.nih.gov/ij/) (Schneider et al., 2012). The image background was adjusted offline using offset control. To measure Itsn1 levels in the calyces, we thresholded vGlut1 images and then used the region of interest (ROI) manager tool (ImageJ) to select foreground pixels. The selected ROIs were used to measure the offset-corrected mean intensity in respective Itsn1 images. To minimize variability, we quantified fluorescence intensities at equal depths from the slice surface. This approach reduced variations due to antibody penetration and tissue scattering, which could affect excitation and emission.

To analyse the spatial distribution of Itsn1 around VGCCs, we drew straight lines (at least 15 lines per calyx) across the VGCC clusters. We chose VGCC clusters located at the inner edge of A594d-labelled calyces (typically at the largest calyx cross-section) as this orientation helped the optical separation of pre- and postsynaptic compartments in the $x$–$y$ dimension (∼184 and 231 nm resolution at 515 and 648 nm excitation, respectively). Lines were placed at the midpoint of cluster peaks, perpendicular to the edge of A594d labelling. To ensure unbiased cluster selection, only the citrine and A594d channels were turned on while Itsn1 images were made invisible. To minimize variability, 20 pairs of calyces from different age groups were quantified from the fluorescence intensity at equal depths from the slice surface. Brightness and contrast adjustments were made for presentation purposes.

## Data analysis

Data were analysed offline with the pCLAMP 10 software package (Molecular Devices) and Excel (Microsoft). Curve fittings were performed using Clampfit (Molecular Devices). Sample sizes were determined based on previous studies using similar experimental protocols (Fekete et al., 2019; Yang & Wang, 2006; Yang et al., 2014). Statistical analysis was conducted using two-way ANOVA with Tukey's test, Mann–Whitney U test or unpaired Student's $t$ test assuming unequal variances. All analyses were two-tailed tests with a significance threshold set at $P < 0.05$. Data were presented as mean ± standard deviation (SD), with $n$ representing the number of cells or samples obtained from more than three mice per group.

## Results

### Itsn1 is present in the calyx of Held synapse and affects the developmental upregulation of vGlut1

To determine how Itsn1 expression changes at the calyx of Held synapse during development, we microdissected brain tissue from the MNTB in WT and Itsn1-Mut mice, before and after hearing onset. The mutant mice, created using a gene-trap embryonic stem cell line, had a complete loss of function of the *Itsn1* gene, eliminating both short and long isoforms of Itsn1 in the whole brain (Sengar et al., 2013). Western blot analysis of MNTB homogenates revealed comparable levels of Itsn1 protein in immature and mature WT mice (Fig. 1*A* and *B*). The absence of Itsn1 bands in mutant samples confirmed their genotype. To examine Itsn1 location in the MNTB, we stained brainstem slices with antibodies specific for Itsn1 and a presynaptic marker vGlut1. We then acquired confocal images in a pairwise manner and quantified fluorescence intensity at the same depth from the slice surface (Fig. 1*C*). In line with an early report (Sakaba et al., 2013), Itsn1 was present in vGlut1-expressing presynaptic terminals, and in the soma of postsynaptic neurons of WT mice (Fig. 1*D*). The Itsn1 signals were absent at mutant synapses (Fig. 1*C*). Over development, presynaptic Itsn1 levels were down-regulated but remained noticeable in mature WT synapses (Fig. 1*E*), while vGlut1 levels increased with age (Fig. 1*F*). This vGlut1 increase was abolished in Itsn1-Mut synapses (Fig 1*F*). As vGlut1 is essential for neural development and glutamate loading into SVs (Wojcik et al., 2004), these results indicate a potential role for Itsn1 in regulating transmitter release.

### Basal synaptic transmission remains unaltered at Itsn1-Mut synapses during development

Given the reduced vGlut1 expression in mature Itsn1-Mut synapses (Fig. 1), we tested the effect of Itsn1 loss on synaptic functions. We recorded spontaneous miniature excitatory postsynaptic currents (mEPSCs) from MNTB neurons in brain slices of WT and Itsn1-Mut mice at different developmental stages (Fig. 2*A* and *C*). Consistent with previous reports (Joshi & Wang, 2002; Joshi et al., 2004; Koike-Tani et al., 2005; Taschenberger & von Gersdorff, 2000), mature (P16–20) synapses showed an increased amplitude and frequency of mEPSCs, but shortened decay time compared to immature (P7–11) synapses, probably due to developmental remodelling in both pre- and postsynaptic elements (Joshi et al., 2007; Kochubey et al., 2016; Yang et al., 2011). However, no significant differences in the amplitude, decay time constant or frequency of mEPSCs were noticed between WT and Mut groups at either age (P7–11, Fig. 2*B*; P16–20, Fig. 2*D*).

Next, we recorded evoked EPSCs by stimulating presynaptic axon bundles with a bipolar electrode. As each postsynaptic neuron is typically innervated by one calyx, these EPSCs occurred in an all-or-none manner (Fig. 3*A* and *C*). Again, mature synapses exhibited a larger amplitude and shorter decay time of EPSCs compared to immature synapses, reflecting a developmental enhancement in release efficacy and a transition from slow- to fast-gating AMPA receptors (Fedchyshyn & Wang, 2005; Joshi et al., 2004; Koike-Tani et al., 2005; Yang et al., 2011). Quantitative analysis did not detect any significant differences in the amplitude or kinetics of EPSCs between WT and Mut synapses at either developmental stage (P7–11, Fig. 3*A*; P16–20, Fig. 3*C*).

Lastly, we recorded EPSCs elicited by train stimulation at 100 Hz (Fig. 3*B*) or 300 Hz (Fig. 3*D*). A lower frequency (100 Hz) was chosen for immature synapses as they are unable to respond reliably to high-frequency stimuli (e.g. 300 Hz) due to the long refractory period of wide APs (von Gersdorff & Borst, 2002). When we plotted the normalized EPSC amplitude (relative to the first EPSC in each train) against stimulus time, we noticed a use-dependent reduction in amplitude (i.e. STD), following a single exponential function (Fig. 3*B* and *D*). At both ages, WT and Itsn1-Mut synapses displayed similar rates of STD. As STD mainly results from the depletion of SVs in the RRP (von Gersdorff & Borst, 2002), we estimated RRP size using the same protocols. No significant differences were found between WT and Mut groups at P7–11 (Fig. 3*B*) or P16–20 (Fig. 3*D*), despite a developmental upregulation of the RRP size at both WT and Mut synapses (P7–11 *vs.* P16–20, $P < 0.001$). Taken together, these results suggest that Itsn1 has little effect on unitary or synchronized vesicular release, short-term synaptic plasticity, or RRP size throughout development.

### Age-dependent effect of Itsn1 on the recovery from STD

Previous work on the calyx of Held and other synapses has demonstrated that STD is largely caused by depletion of the RRP, and recovery from STD depends on RRP replenishment (von Gersdorff et al., 1997; Wang & Kaczmarek, 1998). To test if Itsn1 affects RRP replenishment, we employed a dual-train stimulation paradigm. The first train aimed to deplete the RRP, followed by a second train at increasing intervals to assess recovery from depletion. Because immature synapses are unreliable in response to high-frequency stimulation (Joshi et al., 2004, 2007), we used 100 Hz trains for these synapses, while mature synapses were stimulated at 300 Hz in accordance with their large RRP size (Fig. 3). Figure 4*A* and *C* present such EPSC recordings from WT and Itsn1-Mut synapses at different

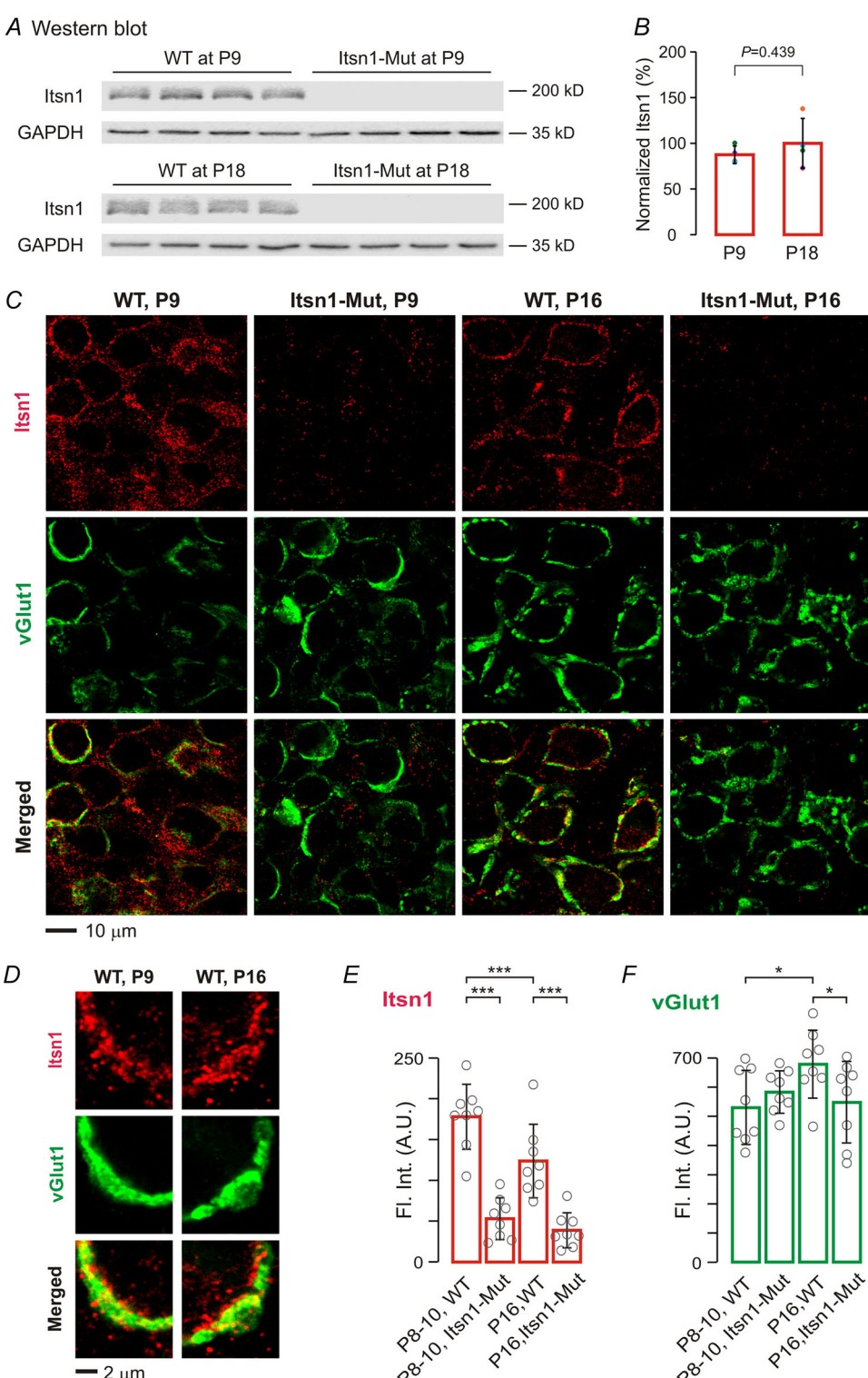

**Figure 1. Itsn1 is expressed in the immature and mature calyx of Held synapses**

*A*, western blots of Itsn1 from MNTB homogenates of WT (*n* = 4) and Itsn1-Mut (*n* = 4) mice at P9 and P18. Itsn1 was undetectable in the mutant tissue. GAPDH was used as a loading control. *B*, normalized Itsn1 protein levels to the average value of P18 WT group (*n* = 4 for each group). *C*, confocal images of MNTB slices stained for Itsn1 (red) and vGlut1 (green) from WT and Itsn1-Mut mice at P9 and P16. Note the absence of Itsn1 labelling in mutant groups. *D*, high-resolution images of Itsn1 and vGlut1 labelling in a P9 and a P16 WT synapse. Itsn1 was present in both pre- and postsynaptic compartments. *E* and *F*, summary plots of the fluorescence intensity (Fl. Int.) for Itsn1 (*E*) and vGlut1 (*F*) labelling in WT and Itsn1-Mut synapses at different ages (*n* = 8 for each group).

Data are presented as mean ± SD. *P*-values are derived from two-way ANOVA with Tukey's tests. Asterisks indicate significant differences across genotypes or developmental stages. *$P < 0.05$; ***$P < 0.001$.

ages. We analysed the replenishment rate by subtracting the average amplitude of the last 4/5 EPSCs in the second train from each EPSC amplitude and dividing the total amplitude of the second train by that of the first train to calculate a recovery percentage. Recovery kinetics was fitted using a two-component exponential cumulative distribution function $f(t) = p_{fast} (1 - e^{-t fast/\tau}) + p_{slow}$ $(1 - e^{-t slow/\tau}) + C$, where $\tau$ represents the time constant and p the weight of each component. For immature synapses, Itsn1 deletion did not alter the recovery time course ($\tau_{fast}$: 0.15 ± 0.08 s for WT, 0.17 ± 0.19 s for Mut, $P = 0.841$; $p_{fast}$: 25.9 ± 8.63% for WT, 30.4 ± 12.6% for Mut, $P = 0.400$; $\tau_{slow}$: 3.98 ± 1.39 s for WT, 4.12 ± 0.83 s for Mut, $P = 0.799$; $p_{slow}$: 71.1 ± 13.4% for WT, 74.6 ± 17.0% for Mut, $P = 0.641$; $n = 10$ for WT, $n = 8$ for Mut; Fig. 4*B*). In contrast, mature Itsn1-Mut synapses showed a significant reduction in the proportion of fast component ($\tau_{fast}$: 0.17 ± 0.17 s for WT, 0.15 ± 0.08 s for Mut, $P = 0.738$; $p_{fast}$: 45.3 ± 16.0% for WT, 23.6 ± 12.6% for Mut, $P = 0.015$) and consequently an increase in the slow component of recovery ($\tau_{slow}$: 1.46 ± 0.51 s for WT, 2.68 ± 0.45 s for Mut, $P < 0.001$; $p_{slow}$: 53.5 ± 11.1% for WT, 74.0 ± 9.53% for Mut, $P = 0.003$) compared to age-matched WT synapses ($n = 8$ for WT, $n = 6$ for Mut; Fig. 4*D*). Collectively, these findings suggest that Itsn1 is crucial for activity-dependent RRP replenishment in mature synapses but not in immature ones.

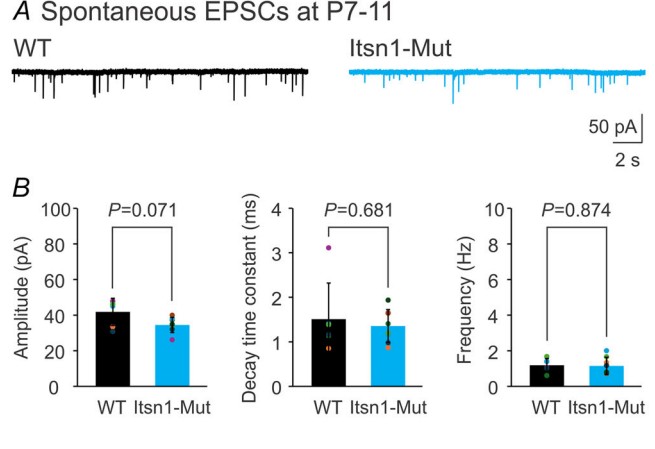

*A* Spontaneous EPSCs at P7–11

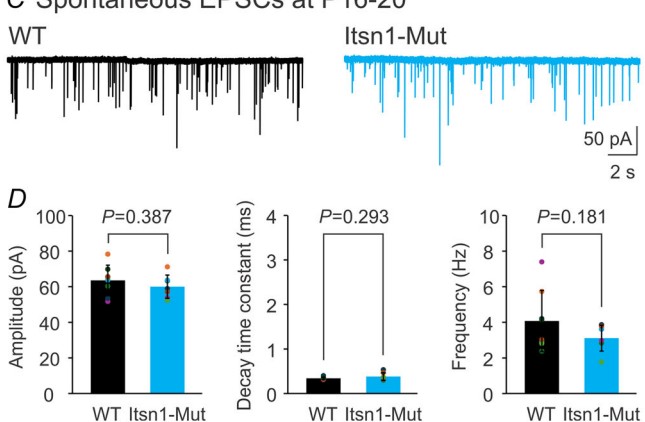

*C* Spontaneous EPSCs at P16–20

**Figure 2. Itsn1 deletion does not affect spontaneous transmitter release at the calyx of Held synapse during development**

*A*, representative traces of spontaneous mEPSCs recorded at −60 mV from P7–11 WT and Itsn1-Mut synapses. *B*, summary plots of mEPSC amplitude (left), decay time constant (middle) and frequency (right) for P7–11 WT ($n = 6$, black) and Itsn1-Mut ($n = 8$, cyan) synapses. Decay time constant was measured by fitting the falling phase of averaged mEPSC for each neuron using an exponential function $f(t) = Ae^{-t/\tau} + C$. Frequency was calculated from the average inter-event interval for each neuron. *C*, examples of spontaneous mEPSCs recorded from P16–20 WT and Itsn1-Mut synapses. *D*, same analysis on mEPSC amplitude (left), decay time constant (middle) and frequency (right) for P16–20 WT ($n = 8$, black) and Itsn1-Mut ($n = 7$, cyan) synapses. Data are presented as mean ± SD. *P*-values are derived from unpaired Student's *t* tests assuming unequal variances.

### Itsn1 mediates Ca²⁺-dependent fast replenishment of the RRP

As the Itsn1-Mut phenotype becomes evident after the onset of hearing (Fig. 4), we thereafter focused on mature synapses. We asked how Itsn1 loss affected the fast and slow components of STD recovery, which are likely to be mediated by distinct mechanisms (Neher, 2010). Since the fast recovery component is $Ca^{2+}$ dependent (Wang & Kaczmarek, 1998), we tested if Itsn1 was involved in this process by injecting EGTA (10 mM), a $Ca^{2+}$ chelator, into WT and Itsn1-Mut terminals. As depicted in Fig. 5*A*, we recorded axonally evoked EPSCs from the postsynaptic neuron with a patch pipette containing 10 mM EGTA attached to the presynaptic calyx. After obtaining baseline EPSCs, we ruptured the presynaptic membrane to allow EGTA diffusion into the calyx for 3 min. Once EGTA equilibrated, the presynaptic pipette was retracted. We then measured EPSCs evoked by a single stimulus or the dual-train stimulation protocol. As expected (Wang & Kaczmarek, 1998), EGTA decreased the EPSC amplitude by 20.3 ± 13.5% for WT ($n = 5$) and 15.4 ± 12.2% for Mut ($n = 9$) ($P = 0.525$; Fig. 5*B*). In addition, EGTA significantly slowed the recovery from STD (Fig. 5*C*), particularly abolishing the fast component in both WT and Mut groups (Fig. 5*D*). The impact of EGTA was more pronounced in WT than in Mut synapses, eliminating group differences in the recovery dynamics ($\tau_{slow}$: 3.20 ± 2.48 s for WT, 2.30 ± 0.41 s for Mut,

$P = 0.464$; $p_{slow}$: 91.9 ± 8.74% for WT, 90.9 ± 6.65% for Mut, $P = 0.853$; $n = 5$ for each group; Fig. 5*D*). These results suggest that Itsn1 plays a specific role in $Ca^{2+}$-dependent fast replenishment of the RRP, but not in the $Ca^{2+}$-independent slow component.

## Fast RRP replenishment requires the Itsn1 DH–PH domains but not dynamin

The DH–PH domains from Itsn1 are known to directly activate Cdc42, a key regulator of actin dynamics that is essential for exocytosis and endocytosis (Hussain

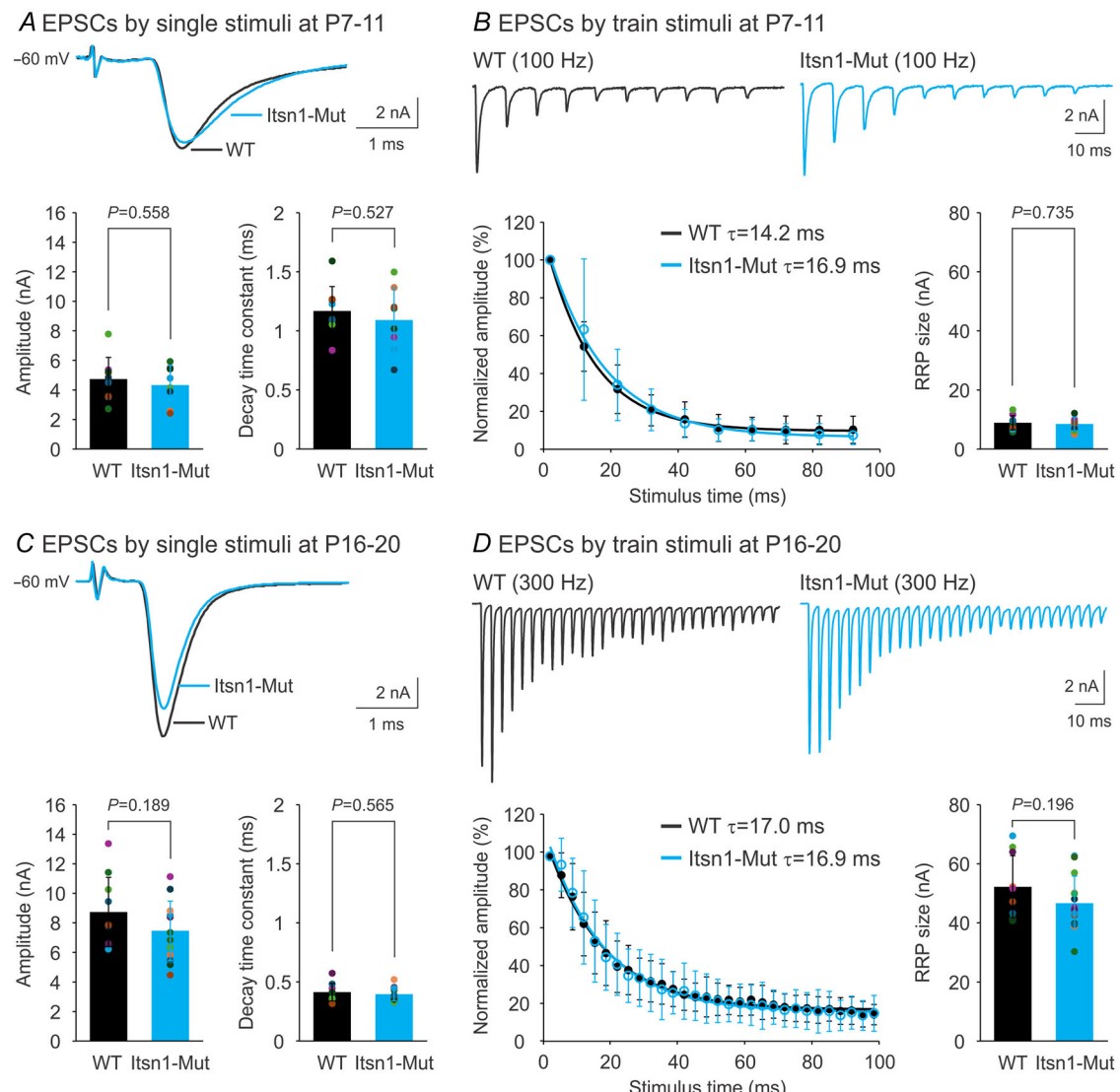

**Figure 3. Itsn1 deletion does not affect evoked transmitter release and short-term plasticity at the calyx of Held synapse during development**

*A*, representative traces of EPSCs from P7–11 WT and Itsn1-Mut synapses evoked by single axonal stimulation (top). Bottom panels summarize the amplitude and decay time constant of these EPSCs for WT (*n* = 9, black) and Itsn1-Mut (*n* = 8, cyan) groups. Decay time constant was measured by fitting the falling phase of the averaged EPSC for each neuron using an exponential function $f(t) = Ae^{-t/\tau} + C$. *B*, representative traces of EPSCs evoked by a 100 Hz, 100 ms train of stimuli in P7–11 WT and Itsn1-Mut synapses (top). Bottom panels summarize the normalized amplitude of these EPSCs to the first response in the train and the RRP size for WT (*n* = 9, black) and Itsn1-Mut (*n* = 8, cyan) groups. Decline of EPSC amplitude over stimuli was fitted with an exponential function $f(t) = Ae^{-t/\tau} + C$. The RRP estimation is described in the Methods. *C* and *D*, examples of EPSCs from P16–20 WT and Itsn1-Mut synapses evoked by a single (*C*) or 300 Hz, 100 ms train of stimuli (*D*). The same analysis on EPSC amplitude and decay time constant for P16–20 WT (*n* = 10, black) and Itsn1-Mut (*n* = 13, cyan) synapses are summarized in *C*. The same analysis on EPSC amplitude decline over stimuli and RRP size for P16–20 WT (*n* = 10, black) and Itsn1-Mut (*n* = 14, cyan) synapses are summarized in *D*. Data are presented as mean ± SD. *P*-values are derived from unpaired Student's *t* tests assuming unequal variances.

et al., 2001). We hypothesized these domains might be crucial for replenishing the RRP following exhaustive exocytosis during repetitive stimulation. To test this, we synthesized a fragment containing the full-length DH–PH domains and delivered it (5 μM) into Itsn1-Mut calyces using patch pipettes. Recovery from STD was then measured by recording EPSCs evoked by two stimulus trains separated by various intervals (Fig. 6A). This intervention noticeably sped up recovery by enhancing the fast recovery component ($p_{fast} = 54.9 \pm 24.3\%$, $n = 6$, $P = 0.025$, compared to the Mut baseline; Fig. 6B).

Concurrently, it reduced the slow recovery component ($p_{slow} = 50.0 \pm 18.4\%$, $n = 6$, $P = 0.023$, compared to the Mut baseline; Fig. 6B), bringing these parameters closer to those observed in WT synapses ($P = 0.426$ for $p_{fast}$, $P = 0.688$ for $p_{slow}$, $n = 6$; Fig. 6B). This result indicates that the DH–PH domain cassette is important for facilitating rapid recovery from STD.

Another intriguing region in Itsn1 is the SH3 domain, which can interact with dynamin, thereby promoting clathrin-mediated endocytosis (Sengar et al., 1999). To examine the impact of Itsn1 deletion on dynamin

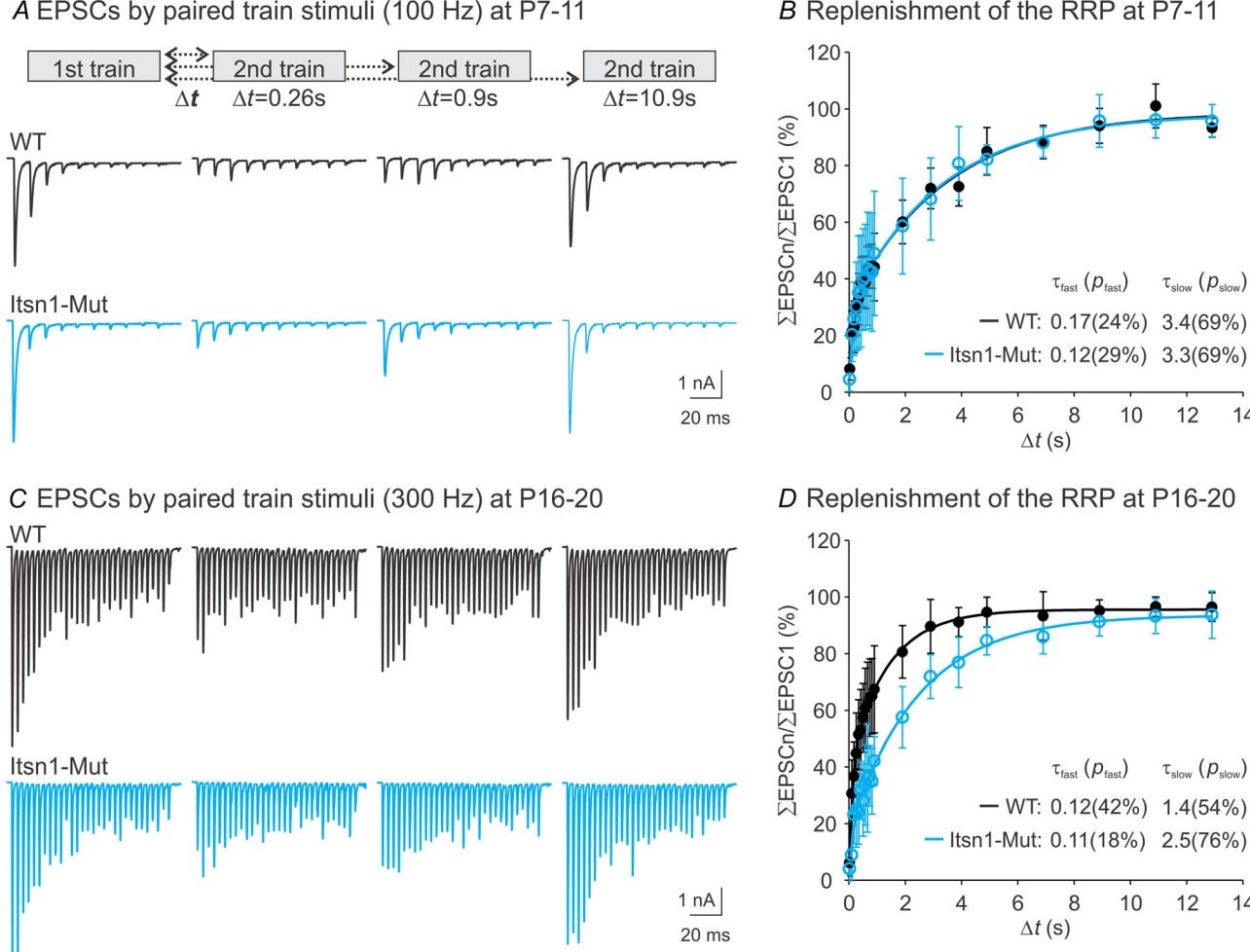

**Figure 4. Itsn1 deletion impairs recovery from STD at mature, but not immature, calyx of Held synapses**
*A*, examples of EPSCs evoked by two separate stimulus trains (100 Hz, 100 ms) at varying intervals (Δ*t*; top panel) from P7–11 WT and Itsn1-Mut synapses. *B*, summary plots showing the recovery from STD induced by depletion of the RRP during train stimulation in P7–11 WT ($n = 10$, black) and Itsn1-Mut ($n = 8$, cyan) synapses. The replenishment rate was analysed by subtracting the average amplitude of the last four EPSCs in the second train from each EPSC amplitude. The total amplitude of the second train was then normalized to that of the first train to obtain a recovery percentage. The recovery time course was fitted using a two-component exponential cumulative distribution function $f(t) = p_1(1 - e^{-t1/\tau}) + p_2(1 - e^{-t2/\tau}) + C$. Time constant τ (s) and weight *p* (%) of each component are specified. *C*, examples of EPSCs evoked by two stimulus trains (300 Hz, 100 ms) separated by various intervals from P16–20 WT and Itsn1-Mut synapses. *D*, the same analysis on the recovery dynamics for P16–20 WT ($n = 8$, black) and Itsn1-Mut ($n = 6$, cyan) synapses, except that the average amplitude of the last five EPSCs in the second train was used for subtraction. Data are presented as mean ± SD. Continuous lines represent fits to the mathematical model.

expression, we performed western blot analysis on MNTB lysates from WT and Itsn1-Mut mice at P18 (Fig. 6C). No difference in dynamin levels was found between WT and Mut groups (Fig. 6D), implying that Itsn1 deletion does not alter overall dynamin expression at mature calyx of Held synapses. Knowing that Itsn1 disruption impairs SV endocytosis via its interaction with dynamin (Sengar et al., 1999; Winther et al., 2013), and dynamin blockade prevents fast recovery from STD in the immature calyx of Held synapses (Sakaba et al., 2013), we postulated that interference with dynamin function might recapitulate the Itsn1-Mut phenotype. Surprisingly, acute inhibition of dynamin activity in WT synapses with dynasore (100 μM) or Dynole 34-2 (10 μM) did not affect recovery from STD, as evidenced by analysis of the fast and slow recovery

components (dynasore: $p_{fast} = 53.3 \pm 13.3\%$, $P = 0.355$; $p_{slow} = 49.9 \pm 13.2\%$, $P = 0.630$; $n = 5$, compared to the WT baseline; Dynole 34-2: $p_{fast} = 45.9 \pm 12.6\%$, $P = 0.950$; $p_{slow} = 53.0 \pm 12.0\%$, $P = 0.942$; $n = 5$, compared to the WT baseline; Fig. 6E and F). These data indicate that Itsn1 mediates RRP replenishment, independent of dynamin.

### Presynaptic APs and Ca²⁺ influx through VGCCs are intact at mature Itsn1-Mut synapses

To determine if the Itsn1-Mut phenotype resulted from alterations in presynaptic inputs, we made current-clamp recordings of APs from mature calyces following axonal stimulation (Fig. 7A). We found no significant differences

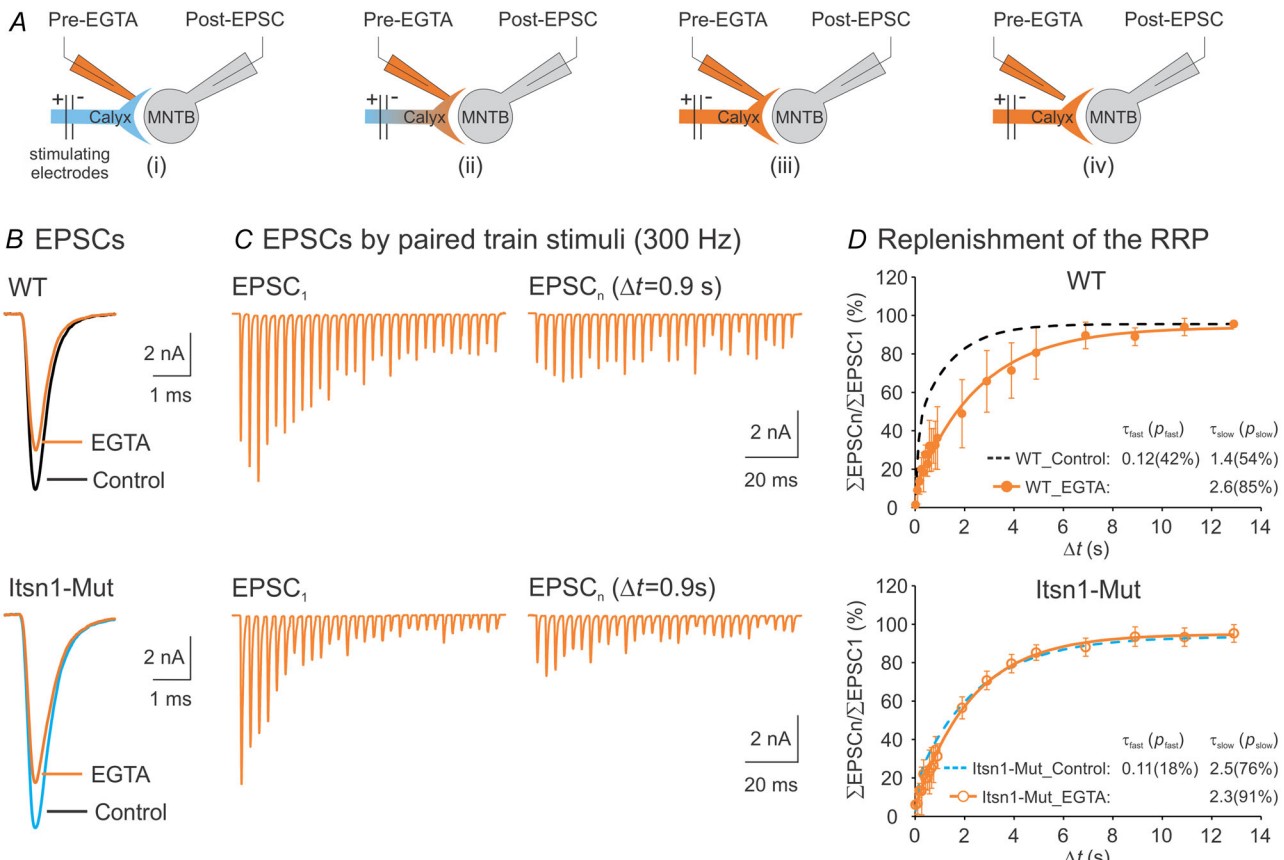

**Figure 5. Itsn1 deletion specifically targets Ca²⁺-dependent recovery from STD at mature calyx of Held synapses**
*A, schematics depicting the process of injecting EGTA (10 mM), a Ca²⁺ chelator, into a calyx nerve terminal while recording EPSC from a postsynaptic MNTB neuron: (i) EPSC recording in the presynaptic cell-attached mode; (ii) EGTA infusion into the terminal; (iii) EGTA equilibrium after 3 min of loading; (iv) EPSC recording after removing the presynaptic electrode. B, examples of EPSCs before and after EGTA injection into P16–20 WT (top) and Itsn1-Mut (bottom) synapses. C, EPSCs elicited by two stimulus trains (300 Hz, 100 ms) separated by a 0.9 s interval from a WT (top) or Itsn1-Mut (bottom) synapse after EGTA injection. D, recovery time courses of WT (n = 5, orange circles, top) and Itsn1-Mut (n = 5, orange circles, bottom) synapses after EGTA infusion were fitted using a one-component exponential cumulative distribution function $f(t) = p(1 - e^{-t/\tau}) + C$. Time constant τ (s) and weight p (%) are specified. For comparison, average recovery curves from age-matched WT (black) and Itsn1-Mut (cyan) mice (as shown in Fig. 4D) were added (dotted lines). Data are presented as mean ± SD. Continuous lines represent fits to the mathematical model.*

in AP amplitude or half-width between WT and Mut synapses (Fig. 7*B*). Based on these measurements, we designed 0.3 ms long AP-like square pulses (−80 to 40 mV). To mimic the train stimulation used in assessing RRP replenishment (Fig. 4), we generated two-pulse trains (300 Hz, 100 ms), separated by a 0.1 s inter-val, which falls within the timescale of the fast recovery component. In the presence of tetrodotoxin (0.5 μM), TEA (10 mM) and 4-aminopyridine (0.3 mM) to block $Na^+$ and $K^+$ channels, we recorded presynaptic $Ca^{2+}$ currents ($I_{Ca}$) from WT (middle panel) and Itsn1-Mut (bottom panel) synapses responding to these train stimuli

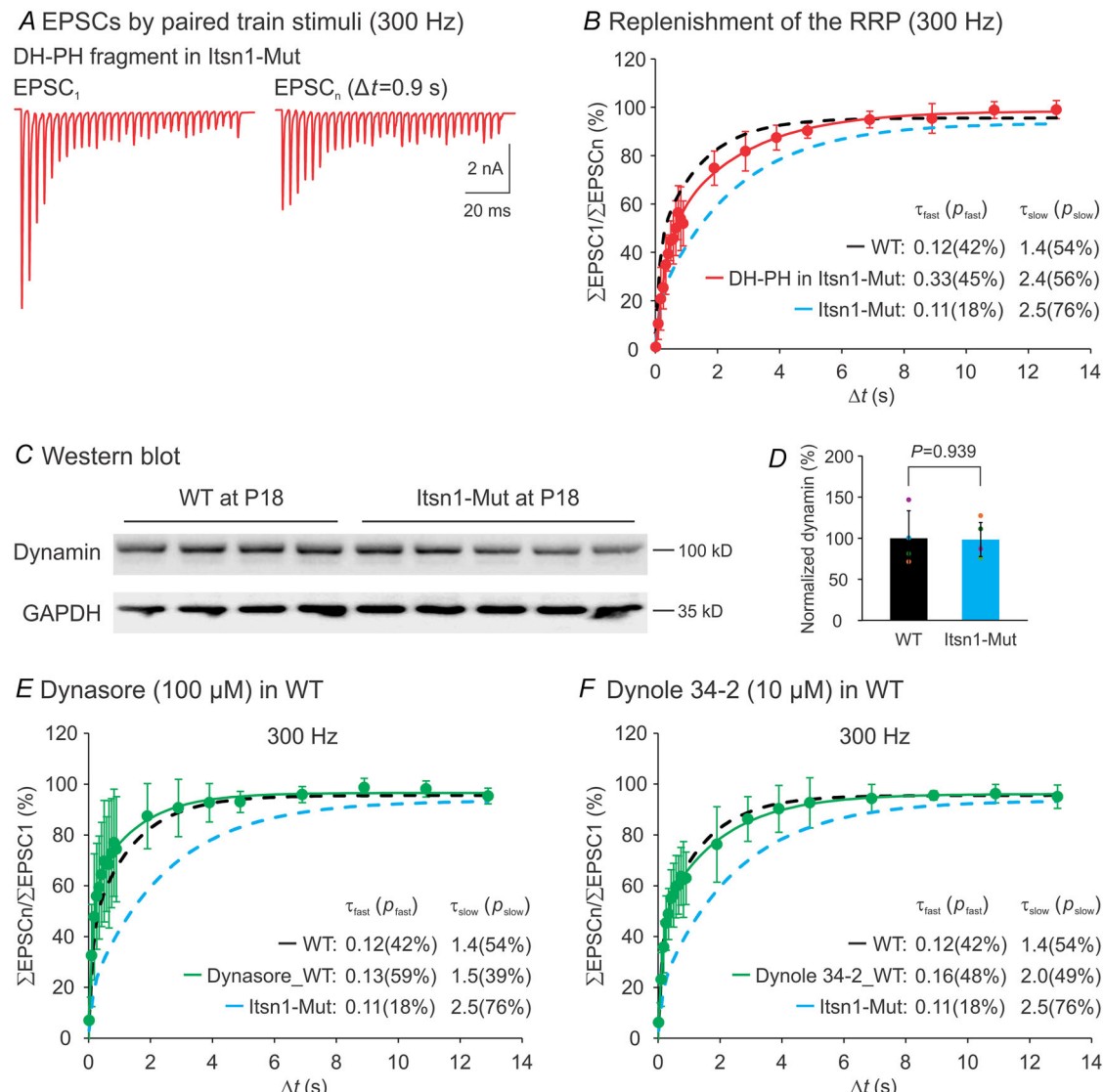

**Figure 6. Itsn1 mediates fast recovery from STD through its DH–PH domains, independent of dynamin, at mature calyx of Held synapses**

*A*, examples of EPSCs evoked by two stimulus trains (300 Hz, 100 ms) separated by a 0.9 s interval after DH–PH fragment infusion (5 μM) into the nerve terminal of P16–20 Itsn1-Mut synapses. *B*, summary plot of recovery from STD in this condition ($n = 6$, red circles). The recovery progression was fitted using a two-component exponential cumulative distribution function $f(t) = p_1(1 - e^{-t1/\tau}) + p_2(1 - e^{-t2/\tau}) + C$. Time constant $\tau$ (s) and weight $p$ (%) of each component are indicated. *C*, western blot analysis of dynamin in MNTB homogenates from WT ($n = 4$) and Itsn1-Mut ($n = 5$) mice at P18. GAPDH was used as a loading control. *D*, normalized dynamin levels to the average value of the WT group. *E* and *F*, summary plots of recovery from STD after extracellular perfusion with dynamin inhibitors, dynasore (100 μM, *E*) or dynole 34-2 (10 μM, *F*), in P16–20 WT synapses ($n = 5$ for each group, green circles). The recovery progression was fitted using the same two-component exponential cumulative distribution function. For comparison, average recovery curves from age-matched WT (black) and Itsn1-Mut (cyan) mice (as shown in Fig. 4*D*) were added to the summary plots (dotted lines). Data are presented as mean ± SD. Continuous lines represent fits to the mathematical model.

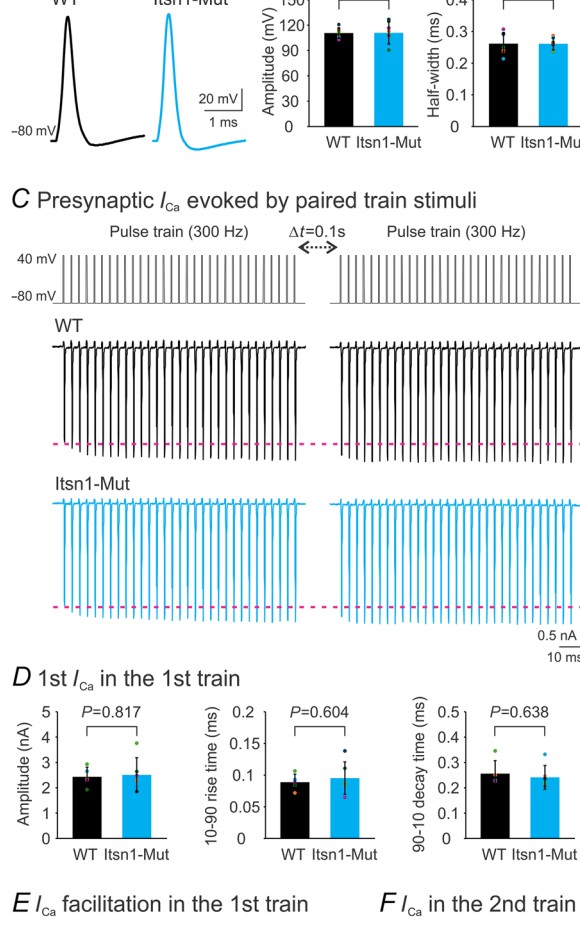

and Itsn1-Mut ($n = 6$) groups are indicated. *F*, normalized amplitude of the 1st $Ca^{2+}$ current in the 2nd train, expressed as a percentage of the 1st $Ca^{2+}$ current in the 1st train, for WT ($n = 5$, black) and Itsn1-Mut ($n = 6$, cyan) groups. Data are presented as mean ± SD. *P*-values are derived from unpaired Student's *t* tests assuming unequal variances.

(top panel; Fig. 7*C*). There were no differences in the amplitude and kinetics of the initial $I_{Ca}$ during the first train between groups (Fig. 7*D*). When normalized to the initial $I_{Ca}$, the amplitude of $I_{Ca}$ increased exponentially over time during the first train (Fig. 7*E*), displaying characteristic activity-dependent facilitation of VGCCs (Nanou & Catterall, 2018). However, no group difference was seen in the facilitation profile. This $I_{Ca}$ facilitation persisted into the second train (Fig. 7*C*), and the levels of facilitation were similar between groups, as shown by the normalized amplitude of the initial $I_{Ca}$ in the second train to that in the first train (Fig. 7*F*). Our results suggest the absence of Itsn1 does not affect the presynaptic AP waveform or $Ca^{2+}$ entry through VGCCs. The effect of Itsn1 on STD recovery probably arises from its role in organizing downstream synaptic events.

### Itsn1 clusters closer to VGCCs as development proceeds

Immunostaining experiments revealed a lower level of Itsn1 in mature WT calyces (Fig. 1*E*), where $Ca^{2+}$-dependent replenishment of the RRP was robust (Fig. 4*C* and). These observations led us to hypothesize that subsynaptic reorganization of Itsn1 during development might underlie the Itsn1-mediated phenotype. Itsn1 is known to facilitate SV replenishment at release sites (Sakaba et al., 2013). We thus examined the spatial relationship between Itsn1 and VGCCs in nerve terminals using *Cacna1a^Citrine* mice, in which the N-terminus of the P/Q-type VGCC $\alpha 1$ subunit was tagged with citrine, a GFP variant (Mark et al., 2011). Previously we have detailed the number, size and distribution of VGCC clusters in the calyces (Fekete et al., 2019). We loaded calyces with Alexa Fluor 594 dextran (A594d), a morphological tracer, and immunolabelled Itsn1 (Fig. 8*A*). We then imaged and analysed immature and mature calyces in a pairwise manner. Because Itsn1 is present at both pre- and postsynaptic sites (Fig. 1), we drew lines across VGCC clusters perpendicular to the edge of the A594d labelling at the largest cross-section of calyces to separate the pre- and postsynaptic compartments in the *x*–*y* dimension with the least overlap (Fig. 8*A*). The centre of each line (at 0 μm) was placed at the VGCC cluster peak, with the right and left sides representing pre- and postsynaptic compartments, respectively (Fig. 8*B*

**Figure 7. Itsn1 deletion does not affect presynaptic APs or $Ca^{2+}$ influx through VGCCs at mature calyx of Held synapses**
*A*, representative APs evoked by axonal stimulation in P16–20 WT (left) and Itsn1-Mut (right) calyces. *B*, summary plots comparing AP amplitude and half-width between WT ($n = 7$, black) and Itsn1-Mut ($n = 7$, cyan) groups. *C*, $Ca^{2+}$ currents elicited in a WT calyx (middle) or an Itsn1-Mut calyx (bottom) by two 300 Hz 100 ms pulse trains (each pulse: −80 to 40 mV, 0.4 ms duration), separated by a 0.1 s interval (top). Extracellular $Ca^{2+}$ concentration was 2 mM. Tetrodotoxin (0.5 μM), TEA (10 mM) and 4-aminopyridine (0.3 mM) were used to inhibit $Na^+$ and $K^+$ channels. *D*, summary plots of the amplitude, 10–90 rise time and 90–10 decay time of the 1st $Ca^{2+}$ current in the 1st train for WT ($n = 5$, black) and Itsn1-Mut ($n = 6$, cyan) groups. *E*, the amplitude of $Ca^{2+}$ currents, normalized to the 1st $Ca^{2+}$ current of the 1st train, plotted against stimulus time and fitted with a one-component exponential cumulative distribution function $f(t) = p(1 - e^{-t/\tau}) + C$. Time constants $\tau$ (s) for WT ($n = 5$)

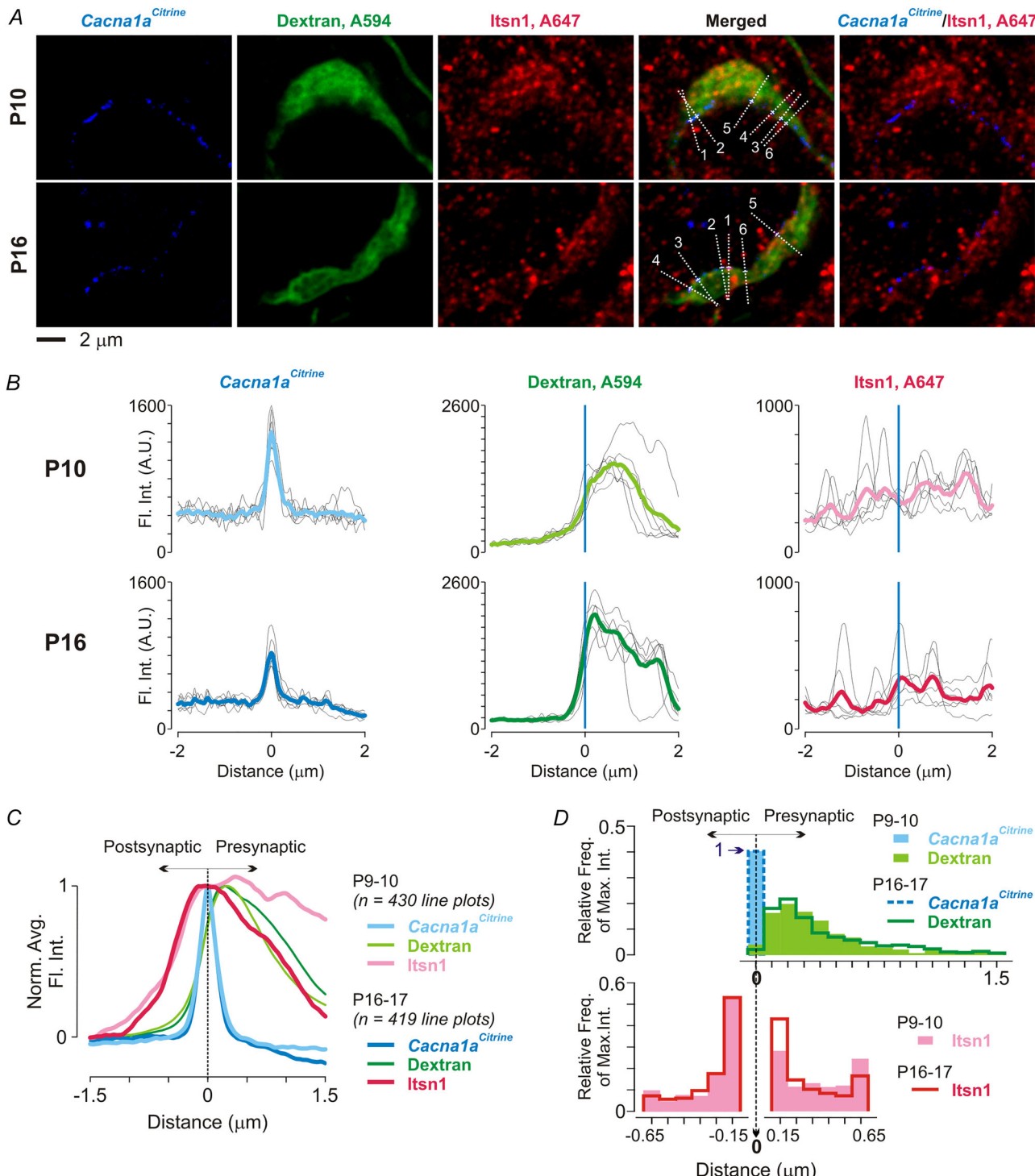

**Figure 8. Subsynaptic enrichment of Itsn1 near VGCCs at the calyx of Held synapse during development**
*A*, confocal images of a P10 (top) and a P16 (bottom) calyx from *Cacna1a^Citrine* mice, in which P/Q-type VGCCs were tagged with citrine (blue). The calyces were filled with A594 dextran (green). Itsn1 immunolabeling was visualized with A647 (red). Dashed lines indicate six examples used for examining subsynaptic distribution of Itsn1. Lines were positioned at the midpoint of VGCC clusters perpendicular to the edge of A594 labelling. *B*, fluorescence intensity (Fl. Int.) along the six individual lines (thin lines) and their averages (thick lines). All line plots were aligned with the peak of *Cacna1a^Citrine* labelling (at 0 μm). *C*, normalized average fluorescence intensity for *Cacna1a^Citrine* (blue), dextran (green) and Itsn1 (red), showing tighter Itsn1 enrichment at VGCC clusters in P16–17 calyces (darker

colours, *n* = 419 line plots from 20 calyces) than in P9–10 calyces (lighter colours, *n* = 430 line plots from 20 calyces). Vertical line at the peak separates pre- and postsynaptic compartments. *D*, relative frequency of maximal intensity for *Cacna1a*^*Citrine* (blue; at 0 μm), dextran (green) and Itsn1 (red), showing Itsn1 maxima were more frequently localized near VGCC clusters in P16–17 calyces than those in P9–10 calyces. Itsn1 maxima in the postsynaptic compartment were similar between groups. Filled areas and darker lines represent P9–10 and P16–17 groups, respectively. Itsn1 maxima were detected either 0.1 to 0.7 μm (presynaptic) or −0.1 to −0.7 μm (postsynaptic) away from the midline, separated by a 0.2 μm window around the peak of VGCC clusters.

and *C*). The presence or absence of A594d signals further validated the pre- and postsynaptic separation. Individual line plots showed a punctate distribution of Itsn1 with varying peak intensities (Fig. 8*B*). There was no difference between groups in the average amount of Itsn1 at the VGCC cluster peaks (P9–10: 294 ± 171, *n* = 430 lines/20 calyces/4 mice; P16–17: 293 ± 154, *n* = 419 lines/20 calyces/4 mice; *P* = 0.511, Mann–Whitney test; Fig. 8*C*). However, there was a significant divergence in Itsn1 levels as the distance from the VGCC clusters increased in the presynaptic compartment (Fig. 8*C*). The Itsn1 level declined faster over distance in mature calyces than in immature ones. The difference became significant at 0.32 μm (P9–10: 303 ± 192, *n* = 430 lines; P16–17: 273 ± 173, *n* = 419 lines; *P* = 0.042, Mann–Whitney test) and was more pronounced at 0.6 μm (P9–10: 292 ± 196, *n* = 430 lines; P16–17: 247 ± 176, *n* = 419 lines; *P* < 0.0001, Mann–Whitney test). The amount of postsynaptic Itsn1 did not differ between groups.

Given the large variability in individual line plots of Itsn1 due to its punctate labelling pattern, we further investigated the spatial distribution of the maxima in these plots. We set a search window from 0.1 to 0.7 μm for the presynaptic compartment and −0.7 to −0.1 μm for the postsynaptic compartment (Fig. 8*D*). We avoided a 0.2 μm window around the VGCC peaks (−0.1 to 0.1 μm) where the pre- and postsynaptic signals mostly overlapped due to the limit of optical resolution. Analysis of the maximal intensity distribution for VGCC clusters, A594d and Itsn1 showed that Itsn1 maxima in mature calyces were more frequently localized near VGCC clusters compared to immature calyces (Kolmogorov–Smirnov *D* = 0.193, *P* < 0.0001). In contrast, the spatial distribution of Itsn1 relative to VGCCs in the postsynaptic compartment was similar between the age groups (Kolmogorov–Smirnov *D* = 0.041, *P* = 0.865). These data indicate that during development, Itsn1 and VGCCs become spatially closer, which may underlie the age-dependent regulation of Itsn1 in RRP replenishment.

### Itsn1 is essential for high-fidelity neurotransmission at mature calyx of Held synapses

To probe the physiological role of Itsn1 in neurotransmission across the calyx of Held synapse, we evaluated the efficacy of excitation–spike coupling, which depends on both presynaptic glutamate release to generate graded depolarization and the intrinsic excitability of postsynaptic neurons (Purves et al., 2019). Given the abundant Itsn1 expression in MNTB neurons (Figs 1 and 8), we assessed their intrinsic excitability after blocking excitatory inputs with AMPA and NMDA receptor antagonists, NBQX (1 μM) and APV (50 μM), and inhibitory inputs with GABA and glycine receptor antagonists, bicuculline (10 μM) and strychnine (1 μM). Current steps were applied to elicit spikes from MNTB neurons, as illustrated in Fig. 9*A*. Steady-state potentials, measured at the end of each step, rendered the input–output relationship, which followed an exponential function (Fig. 9*B*, left panel). The resulting curves for WT and Itsn1-Mut groups overlapped, indicating that the input resistance of these neurons was the same. When we analysed the number of spikes evoked by each current step using a Boltzmann function (Fig. 9*B*, right panel), there was no difference in the theoretical maximum number of spikes between WT and Mut groups (12.5 ± 3.73 for WT, *n* = 8; 11.3 ± 2.09 for Mut, *n* = 16; *P* = 0.780), suggesting that Itsn1 deletion does not affect the intrinsic excitability of postsynaptic neurons.

We next recorded APs from MNTB neurons while stimulating presynaptic axons with two stimulus trains (300 Hz, 100 ms), separated at various intervals (Fig. 9*C*, top panel). We found that APs triggered by the first train were comparable between WT (Fig. 9*C*, middle panel) and Itsn1-Mut synapses (Fig. 9*C*, bottom panel). Quantitative analysis of the initial APs in the first train revealed no group differences in their amplitude, area integral, rise time or decay time (Fig. 9*D*). This is corroborated by the observation that Itsn1 deletion did not change EPSCs evoked by single or train stimulation (Fig. 3). In contrast, we found more AP failures responding to the second train in Itsn1-Mut synapses, particularly at short inter-train intervals, such as 0.1 s (Fig. 9*E*). AP failures gradually disappeared when the intervals were prolonged. Additionally, Itsn1-Mut synapses had a longer rise time for the initial AP during the second train than WT ones (Fig. 9*C*), indicating a diminished synaptic drive due to Itsn1 deletion. By plotting the normalized rise time of the initial AP in the second train to that in the first train, we noticed a significant increase in the rise time for the Mut group compared to the WT group when the inter-train intervals were shorter than 1 s (Fig. 9*F*). The group difference vanished at longer intervals. These results further support our view that Itsn1 deletion specifically

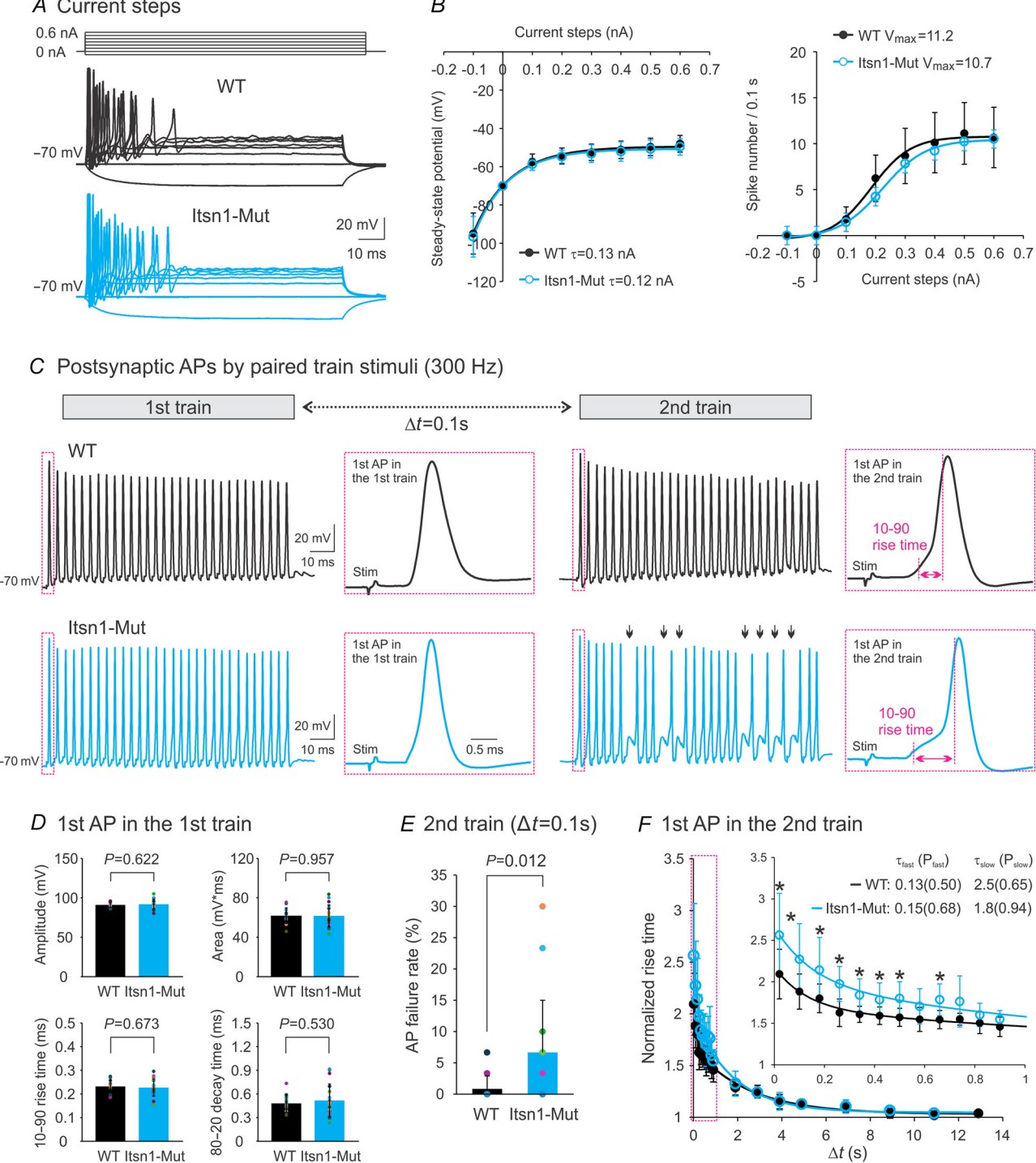

**Figure 9. Itsn1 deletion impairs the fidelity of neurotransmission at mature calyx of Held synapses without affecting the intrinsic excitability of postsynaptic neurons**

A, typical spikes from MNTB neurons in P16–20 WT (middle) and Itsn1-Mut (bottom) synapses evoked by current steps from −0.1 to 0.6 nA over 100 ms (top). Synaptic inputs were blocked by bicuculline (10 μM), strychnine (1 μM), NBQX (1 μM) and APV (50 μM). B, steady-state potentials measured within the last 5 ms of current steps (left) and numbers of spikes (right) for WT (n = 10, black circles) and Itsn1-Mut (n = 18, cyan circles) neurons. Left panel includes fits to a single exponential function $f(t) = Ae^{-t/\tau} + C$ (continuous lines). Right panel includes fits to a charge–voltage Boltzmann function $f(I) = V_{max}/(1 + e^{(Imid-I)/Ic}) + C$ (continuous lines), where $V_{max}$ is the theoretical

maximum number of spikes, $I_{mid}$ is the depolarization current needed to elicit half of the maximum number of spikes and $I_c$ is the steepness of the Boltzmann curve. C, postsynaptic APs triggered by axonal stimulation with two trains (300 Hz, 100 ms) separated by 0.1 s (top) from a WT (middle) and an Itsn1-Mut (bottom) synapse. Arrows indicate spike failures in response to the stimuli. The 1st APs in each train are highlighted to show the rise time difference between groups. D, comparisons of amplitude, area integral, rise time and decay time of the 1st AP in the 1st train between WT ($n = 13$, black) and Itsn1-Mut ($n = 18$, cyan) groups. E, AP failure rates in the 2nd train, delivered 0.1 s after the 1st train, for WT ($n = 8$, black) and Itsn1-Mut ($n = 16$, cyan) groups. A failure rate was calculated as the percentage of failures relative to the total stimulus number. F, normalized rise time of the 1st AP in the 2nd train to that in the 1st train at various inter-train intervals for WT ($n = 9$, black circles) and Itsn1-Mut ($n = 9$, cyan circles) groups. Continuous lines is fit to a two-component exponential cumulative distribution function $f(t) = p_1(1 - e^{-t1/\tau}) + p_2(1 - e^{-t2/\tau}) + C$. Time constant $\tau$ (s) and weight $p$ (%) of each component are indicated. A zoomed-in graph shows prolonged rise time of the 1st AP in the 2nd train at Itsn1-Mut synapses compared to WT synapses. Data are presented as mean ± SD. P-values are derived from unpaired Student's $t$ tests assuming unequal variances.

impairs the fast component of recovery from STD, thereby accounting for spike failures and delayed onset of the first APs in the second train.

## Discussion

Our study demonstrates an age-dependent role for Itsn1 in the recovery from STD, which is absent in immature synapses but becomes prominent in mature ones during high-frequency activity. This Itsn1-mediated fast recovery is driven by $Ca^{2+}$-dependent replenishment of the RRP, as shown by the blunted effect of EGTA in mature Itsn1-Mut synapses. Itsn1 facilitates the fast replenishment, probably by regulating actin dynamics via its DH–PH cassette, independent of dynamin. Surprisingly, Itsn1 protein expression decreases over development but shows sub-synaptic enrichment near P/Q-type VGCCs at mature terminals. This developmental repositioning may allow Itsn1 to effectively respond to local $Ca^{2+}$ changes, accelerating $Ca^{2+}$-dependent RRP replenishment and supporting high-frequency neurotransmission.

At immature synapses, the absence of Itsn1 did not affect recovery from STD induced by physiological stimuli (i.e. APs; Fig. 4), despite Itsn1 being expressed in these terminals (Fig. 1). In synapses of a similar age, Sakaba et al. (2013) applied voltage steps to activate transmitter release from fast- and slow-releasing SVs and showed that Itsn1 deletion impaired $Ca^{2+}$-dependent replenishment of the fast-releasing pool. This discrepancy may stem from the different concepts underlying the RRP and the fast-releasing pool, as well as from the different stimulation protocols. Continuous voltage steps might produce greater $Ca^{2+}$ accumulation than AP bursts, thus facilitating the engagement of Itsn1 in SV replenishment at immature synapses. Given the broader APs and lower $Ca^{2+}$ buffer and extrusion capacities at this developmental stage (von Gersdorff & Borst, 2002), it is unlikely that residual $Ca^{2+}$ after AP-triggered transmitter release fails to activate $Ca^{2+}$-dependent signalling for refilling the RRP. In immature calyces, P/Q-, N- and R-type VGCCs cooperate to mediate SV fusion while in mature synapses,

only P/Q-type VGCCs mediate the process (Fedchyshyn & Wang, 2005; Iwasaki & Takahashi, 1998). Interestingly, P/Q-type VGCCs are also critical for $Ca^{2+}$-dependent endocytosis (Midorikawa et al., 2014). The low expression of P/Q-type VGCCs in immature synapses may prevent their interaction with Itsn1, impeding $Ca^{2+}$-dependent replenishment of the RRP near release sites. During development, Itsn1 translocates to the proximity of P/Q-type VGCCs (Fig. 8), probably forming specialized $Ca^{2+}$ domains that expedite the entry of SVs to the RRP or the restoration of release sites following exocytosis (Neher, 2010).

At mature synapses, multiple functional domains of Itsn1 can contribute to its role in $Ca^{2+}$-dependent recovery from STD (Figs 4 and 5). The C2 domain may act as a $Ca^{2+}$ sensor to initiate SV recruitment and fusion at depleted sites (Gundelfinger et al., 2003; Zhang et al., 2013). As a scaffolding protein, Itsn1 may also play the role by modulating Munc13-1. Previous studies suggest that Munc13-1 activation by calmodulin primes docked SVs and restores release sites, thereby facilitating RRP replenishment (Chen et al., 2013; Lipstein et al., 2013). Itsn1-Mut synapses exhibited a similar deficit in $Ca^{2+}$-dependent replenishment as observed in neurons with mutations that disrupt the C2 domain of Munc13-1 or Munc13-1 interaction with calmodulin (Lipstein et al., 2013; Lipstein et al., 2021). Via its SH3 domain, Itsn1 can bind to the proline-rich domain of vGlut1 or synaptobrevin, potentially tethering vGlut1-loaded SVs before fusion or clearing release sites after fusion (Japel et al., 2020; Zhang et al., 2019). However, dynamin, which interacts with the Itsn1 SH3 domain in endocytosis (Sengar et al., 1999; Wang et al., 2008; Winther et al., 2013), is unlikely to be involved in fast RRP refilling, as demonstrated in our experiments with dynamin inhibitors (Fig. 6). This view is reinforced by the literature showing that SV end-ocytosis at the mature calyx of Held synapse becomes dynamin independent (Yamashita et al., 2010). Instead, Itsn1 probably accelerates RRP replenishment through its DH–PH domains, which function as guanine nucleotide

exchange factors (GEFs) for Cdc42 activation to control actin polymerization (Hussain et al., 2001). This is supported by our observation that the DH–PH fragment effectively rescued the Itsn1-Mut phenotype (Fig. 6), in line with an early report that Itsn1-mediated Cdc42 activity was critical for replenishing fast-releasing SVs at the calyx of Held synapse (Sakaba et al., 2013). Although DH–PH domains are conserved across various proteins (Hoffman & Cerione, 2002), their specificity for each protein depends on sequence variations, 3D structure and binding partners in certain intracellular environments (Ponting & Russell, 2002). While the DH–PH fragment might have effects beyond substituting Itsn1, it probably functioned like Itsn1 in rescuing the specific phenotype related to Itsn1 loss. Collectively, these data support a model where Itsn1 is essential for rapid SV priming or tethering to repopulate the RRP in a $Ca^{2+}$-dependent manner.

Although Itsn1 and Itsn2 are homologues with overlapping tissue expression (Gubar et al., 2013; Herrero-Garcia & O'Bryan, 2017), there is no evidence of functional redundancy between them. In mouse models, Itsn1 deletion leads to learning impairments (Sengar et al., 2013), a reduced midline corpus callosum (Sengar et al., 2013) and synaptic dysfunctions (Gerth et al., 2017; Japel et al., 2020; Sakaba et al., 2013). In contrast, Itsn2-knockout mice do not show these deficits (Gerth et al., 2017; Sengar et al., 2013). The disparity may be due to the enrichment of the Itsn1 long isoform in neurons (Hussain et al., 1999). Thus, the developmental effect of Itsn1 on RRP replenishment (Fig. 4) is unlikely to be caused by a compensatory action of Itsn2, though future studies are required to confirm this.

Itsn1 is present in both pre- and postsynaptic compartments of the calyx of Held synapse throughout development (Figs 1 and 8). However, basal synaptic transmission, including spontaneous and synchronized transmitter release, remains unchanged in Itsn1-lacking synapses (Figs 2 and 3), suggesting that Itsn1 does not affect postsynaptic glutamate receptors mediating these events. In hippocampal neurons, Itsn1 is known to regulate dendritic spine development (Nishimura et al., 2006; Thomas et al., 2009) and long-term plasticity (Jakob et al., 2017), probably by interacting with cell signalling pathways involved in neuronal migration, differentiation and survival (Mintoo et al., 2024). It remains to be determined if Itsn1 plays a role in activity-dependent plasticity at postsynaptic MNTB neurons (Joshi et al., 2007).

In the calyx of Held synapse, SVs are loaded with either vGlut1 or vGlut2, with partially overlapping distribution within the same terminals (Billups, 2005). We noticed a developmental increase in vGlut1, which was abolished by Itsn1 deletion (Fig. 1). While our method could not distinguish whether the vGlut1 increase was due to more vGlut1 per SV or more vGlut1-laden SVs, Itsn1 is clearly necessary for this upregulation. As SVs rarely use both vGlut1 and vGlut2 for glutamate uptake (Upmanyu et al., 2022), the proportion of these two SV populations is probably altered in mature Itsn1-Mut synapses. Despite the fact that vGlut1 enhances SV refilling with glutamate (Nakakubo et al., 2020), its reduced level at Itsn1-Mut synapses might not directly contribute to the slower RRP replenishment, as vGlut1-knockout synapses exhibit faster RRP replenishment (Nakakubo et al., 2020). This indicates that the RRP replenishment is a different process from loading glutamate into SVs, although developmental modulation of the synapse by vGlut1 cannot be excluded. Nevertheless, the deficit in RRP replenishment compromises the precision of excitation–spike coupling, as illustrated in the dual-train stimulation experiment (Fig. 9). Such a paradigm is physiologically relevant to auditory functions, for example gap discrimination (Starr et al., 1991). In summary, we propose that Itsn1 facilitates $Ca^{2+}$-dependent fast replenishment of the RRP by spatial coupling with P/Q-type VGCCs during development to support high-frequency and high-fidelity neurotransmission in the calyx of Held and possibly other synapses.

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

## Additional information

### Data availability statement

All data that support the findings of this study are available upon request.

### Competing interests

None declared.

## Author contributions

L.Y.W., M.W.S. and S.E.E. conceptualized the study. Y.M.Y. and L.Y.W. designed and executed the electrophysiological experiments with assistance from research interns A.L., E.W.S. and A.W., who also started the immunohistochemistry tests. A.F. and G.G. conducted the immunostaining and imaging analysis. M.D.M. and S.H. provided *Cacna1a^{Citrine}* mice. Genotyping and western blot assays were performed by J.A., A.S.S. and J.A. The initial manuscript was drafted by L.Y.W., Y.M.Y. and A.F. All authors reviewed and contributed to the final version of the manuscript.

## Funding

This research was supported by grants from the Canadian Institutes of Health Research (PJT-156034, PJT-156439 and PJT-191780 to L.Y.W.; FDN-154336 to M.W.S.; CIHR-6210100860 to S.E.E.), Natural Science and Engineering Research Council (RGPIN-2017-06665 to L.Y.W.) and Tier 1 Canada Research Chair Program (CRC-95-2324 to L.Y.W.), and a start-up fund from the University of Minnesota (to Y.M.Y.).

## Keywords

action potential, calyx of Held synapse, Intersectin-1, neurotransmission, readily releasable pool

## Supporting information

Additional supporting information can be found online in the Supporting Information section at the end of the HTML view of the article. Supporting information files available:

**Peer Review History**

