## [Peer Review History · The Journal of Physiology]

Intersectin-1 enhances calcium-dependent replenishment of the readily releasable pool of synaptic vesicles over development

Yi-Mei Yang, Adam Fekete, Jason Arsenault, Ameet S Sengar, Jamila Aitoubah, Giovanbattista Grande, Angela Li, Eric W Salter, Alex Wang, Melanie D Mark, Stefan Herlitze, Sean E Egan, Michael W. Salter, and Lu-Yang Wang

DOI: 10.1113/JP286462

Corresponding author(s): Yi-Mei Yang (ymyang@d.umn.edu)

The following individual(s) involved in review of this submission have agreed to reveal their identity: George J. Augustine (Referee #1)

Review Timeline:	Submission Date:	03-May-2024
	Editorial Decision:	03-Jun-2024
	Revision Received:	13-Aug-2024
	Editorial Decision:	30-Aug-2024
	Revision Received:	01-Sep-2024
	Accepted:	06-Sep-2024

Senior Editor: Katalin Toth

Reviewing Editor: Samuel Young

Transaction Report:

Dear Dr Yang,

Re: JP-RP-2024-286462 "Developmental translocation of Intersectin-1 pivots calcium-dependent replenishment of the readily releasable pool of synaptic vesicles" by Yi-Mei Yang, Adam Fekete, Jason Arsenault, Ameet S Sengar, Jamila Aitoubah, Giovanbattista Grande, Angela Li, Eric W Salter, Alex Wang, Melanie D Mark, Stefan Herlitze, Sean E Egan, Michael W. Salter, and Lu-Yang Wang

Thank you for submitting your manuscript to The Journal of Physiology. It has been assessed by a Reviewing Editor and by 2 expert referees and we are pleased to tell you that it is acceptable for publication following satisfactory revision.

LANGUAGE EDITING AND SUPPORT FOR PUBLICATION: If you would like help with English language editing, or other article preparation support, Wiley Editing Services offers expert help, including English Language Editing, as well as translation, manuscript formatting, and figure formatting at www.wileyauthors.com/eoo/preparation. You can also find resources for Preparing Your Article for general guidance about writing and preparing your manuscript at www.wileyauthors.com/eoo/prepresources.

REVISION CHECKLIST:

Please upload two versions of your manuscript text: one with all relevant changes highlighted and one clean version with no changes tracked. The manuscript file should include all tables and figure legends, but each figure/graph should be uploaded as separate, high-resolution files. The journal is now integrated with Wiley's Image Checking service. For further details, see: <https://www.wiley.com/en-us/network/publishing/research-publishing/trending-stories/upholding-image-integrity-wileys-image-screening-service>.

- 'Potential Cover Art' for consideration as the issue's cover image
- Appropriate Supporting Information (video, audio or data set: see https://jp.msubmit.net/cgi-bin/main.plex?form_type=display_requirements#supp)

We look forward to receiving your revised submission.

Yours sincerely,

Katalin Toth
Senior Editor
The Journal of Physiology

REQUIRED ITEMS

- Author photo and profile. First or joint first authors are asked to provide a short biography (no more than 100 words for one author or 150 words in total for joint first authors) and a portrait photograph. These should be uploaded and clearly labelled together in a Word document with the revised version of the manuscript. See Information for Authors for further details.
- You must start the Methods section with a paragraph headed Ethical Approval. A detailed explanation of journal policy and regulations on animal experimentation is given in Principles and standards for reporting animal experiments in The Journal of Physiology and Experimental Physiology by David Grundy *J Physiol*, 593: 2547-2549. doi:10.1113/JP270818). A checklist outlining these requirements and detailing the information that must be provided in the paper can be found at: <https://physoc.onlinelibrary.wiley.com/hub/animal-experiments>. Authors should confirm in their Methods section that their experiments were carried out according to the guidelines laid down by their institution's animal welfare committee, and conform to the principles and regulations as described in the Editorial by Grundy (2015), including an ethics approval reference number. The Methods section must contain a statement about access to food, water and housing, details of the anaesthetic regime: anaesthetic used, dose and route of administration, and method of killing the experimental animals.
- Your manuscript must include a complete Additional Information section, including competing interests; funding; author contributions and acknowledgements.
- Please upload separate high-quality figure files via the submission form.
- Please ensure that the Article File you upload is a Word file.
- Please include an Abstract Figure file, as well as the Figure Legend text within the main article file. The Abstract Figure is a piece of artwork designed to give readers an immediate understanding of the research and should summarise the main conclusions. If possible, the image should be easily 'readable' from left to right or top to bottom. It should show the physiological relevance of the manuscript so readers can assess the importance and content of its findings. Abstract Figures should not merely recapitulate other figures in the manuscript. Please try to keep the diagram as simple as possible and without superfluous information that may distract from the main conclusion(s). Abstract Figures must be provided by authors no later than the revised manuscript stage and should be uploaded as a separate file during online submission labelled as File Type 'Abstract Figure'. Please also ensure that you include the figure legend in the main article file. All Abstract Figures should be created using BioRender. Authors should use The Journal's premium BioRender account to export high-resolution images. Details on how to use and access the premium account are included as part of this email.

EDITOR COMMENTS

Reviewing Editor:

This manuscript is focused on the molecular mechanisms that regulate RRP replenishment. In this manuscript the authors demonstrate that Intersectin 1 is a key regulatory protein in RRP replenishment. Both reviewers found the data important, rigorous, and the findings highly impactful. Both reviewers had specific comments with respect to IHC analysis and correlation to western blot findings. It was pointed out that the western blots show no Intersectin protein, but IHC shows ~50% reduction in protein levels at the calyx. The authors need to address this discrepancy. Furthermore, both commented that the conclusions and interpretations of results are a bit overreaching and conclusions/interpretations should be toned down. In addition, Reviewer#1 has a specific comment with respect to the title. These comments can be addressed by carefully revising and rewriting their manuscript. In addition, please include an Ethics statement in the methods sections as required by Journal of Physiology policy.

Please also see 'Required Items' above.

REFeree COMMENTS

Referee #1:

This paper determines how *Itsn1* regulates synaptic transmission during development. To answer the question, the authors investigated transmitter release at the calyx of Held synapse in young and old mice, both wild-type and *Itsn1*-deficient mutant mice. Remarkably, the physiological consequences of loss of *Itsn1* are quite specific: while virtually every measured presynaptic and postsynaptic property is identical in WT and mutant mice, there is a selective slowing of the fast, Ca²⁺-dependent component of recovery from synaptic depression. Even more remarkable is that this phenotype is observed only in older mice, at an age where the calyx synapse is mature. This developmental switch in the role of *Itsn1* is correlated with an apparent change in the spatial relationship between *Itsn1* and P/Q-type calcium channels. The authors conclude that the colocalization of *Itsn1* and calcium channels is critical for the replenishment of the readily-releasable pool of synaptic vesicles following depletion of this pool during synaptic depression.

The physiological data in the paper are strong and convincing; I am impressed by the specificity of the physiological phenotype. The histological data are OK, as far as they go, and in any case are not the main point of the paper (but see point 1 below).

Here are some suggestions for improvement that the authors should consider:

1. Title: "Developmental translocation of Intersectin-1 pivots calcium-dependent replenishment of the readily releasable pool of synaptic vesicles". I am not sure what "pivots" means in the context of the data shown in the paper. A clearer/more literal title would be advisable, because the authors have not really established any causal relationship between the observed developmental changes in *Itsn1* localization and the *Itsn1*-related changes in recovery from synaptic depression. For example, one could argue that the changes in *Itsn1* localization reflect changes in the location of sites of endocytosis and are unrelated to the location of calcium channels (or sites of exocytosis or rapid RRP replenishment).
2. In Figure 1A-B, including an internal control for the expression level of *Itsn1* would be advantageous to control for possible differences in loading. Reference proteins, such as GAPDH, tubulin, or actin, are typically used. At a minimum, the procedure used for normalization in Fig. 1B should be explained more thoroughly.
3. P. 13, para. 3: "brain tissue containing the medial nucleus of the trapezoid body" - roughly what fraction of the sample does the MNTB represent? I am guessing that it is a pretty minor part of the total sample volume; if so, authors might just want to say "brainstem tissue".
4. Is there any reason to think that *Itsn2* could compensate for the loss of *Itsn1*, particularly in the immature calyx?
5. P. 14, para. 1: Do we know what *Itsn1* is doing in the soma of postsynaptic MNTB neurons? The signal is pretty strong, so it must be doing something.
6. P. 14, line 8: I see no need to list all of the numerical data that are associated with Fig. 1E (and other figures). The whole point of a figure is to visually represent the data, so there is no need for repetition unless some precise numerical comparison is intended. Such lists of extraneous numbers here (and throughout the paper) significantly interrupt the flow for

readers. Also, if the authors really want to leave in the numbers there is no need for 4 significant figures - there is no biological measurement that has an accuracy of 0.01%.

7. Fig. 1E: I do not understand why the fluorescence signals for *Its1n1* are only reduced by 50% in the mutants? Both the Western blots shown in Fig. 1A and the fluorescence images shown in Fig. 1D suggest that these signals should be vanishingly small in the mutants.

8. P. 17, para. 2: Fig. 5A is not really explained in the text.

9. P. 18, para. 2 and Figures 6A and 6B: My only serious scientific concern, in the entire paper, is interpretation of the effects of the DH-PH peptide. Specifically, is this peptide exerting a dominant-negative or gain-of-function effect? The authors clearly interpret their data in terms of the latter possibility, but in fact most small peptides serve as inhibitors of protein-protein interactions, rather than activators. The Hussein et al. (2001) paper used full-length DH-PH (~ 350 amino acids) and saw that this activated Cdc42. Did the authors use the same construct (a fragment or domain, rather than a peptide) or a smaller synthetic peptide (~20-30 amino acids)? I could not find any explicit description of the "peptide" that the authors used or how they synthesized or expressed it. Also, if *Its1n1* works as a Ca²⁺ sensor, as suggested in the Discussion, how would DH-PH rescue in the mutant given that this fragment/peptide lacks the C2 domain of *Its1n1*?

10. Fig. 8D: I am wondering about the resolution of the confocal microscope used for these imaging experiments. The authors state that the confocal microscope used has ~200 nm resolution (p. 13, top), which is consistent with the Abbe diffraction limit (245 nm for 520 nm emission collected with a 1.4 NA objective). However, the *Cacna1a* (Citrine) fluorescence shown in Fig. 8D appears to occupy only a single bin (width 100 nm) and it seems unlikely that the entire distribution of this signal is less than half of the theoretical minimum. Also, note that the resolution should be poorer (approximately 300 nm) for *Its1n1* imaging, because of the longer wavelength of the fluorophore used (A647). Thus, it is possible that the first bin for the "presynaptic" *Its1n1* signal (to the right of the blanked-out bin in the bottom graph of Fig. 8D) still could be coming from the postsynaptic side, which is less than 300 nm away. The bottom line is that the authors should exercise some caution in interpreting the data shown in Fig. 8D.

Finally, here are some very minor suggestions for improving word usage:

p. 3,, last line: "it converts excitatory inputs from globular bushy cells on one side to inhibitory outputs to the superior olive on the other side." Perhaps change "on one side" to "ipsilateral" and "other side" to "contralateral".

p. 4, middle of the page: "alternation" probably should be "alteration"

p. 5, line 7: "clathrin mediated" should be hyphenated

p. 14, lines 1-3: "To assess *Its1n1* location, we co-labeled *Its1n1* with a presynaptic marker vesicular glutamate transporter-1". *Its1n1* was not co-labelled; a second independent label, vGlut1, was used to examine the location of synaptic vesicles.

p. 14, para. 2: "Given that *Its1n1* mutation correlated with reduced vGlut1 at the calyx of Held synapse" this correlation was only observed in older mice, so some rewording seems necessary.

p. 15, para. 2: "As each postsynaptic neuron is typically innervated by one calyx, these EPSCs were large and occurred in an all-or-none manner". Probably should remove (or relocate) the "large" part, because the size of the EPSC is unrelated to innervation by one calyx.

Referee #2:

Synaptic transmission requires reliable exocytosis and endocytosis across a range of stimulation frequencies. At high frequencies of activity, the readily-releasable pool (RRP) of synaptic vesicles becomes depleted and must be rapidly replenished in order to maintain neurotransmission. This study by Yang et al., documents an age-dependent role for intersectin-1 (*Its1n1*) in calcium-dependent recovery from short term depression at the calyx of Held synapse in the auditory brainstem. The authors delineate a number of aspects of synaptic transmission that are unchanged in the *Its1n1* mutants, including calcium entry, action potentials, spontaneous and evoked EPSCs, and RRP size. However, a key finding is that the RRP replenishment after calcium-dependent short term depression is significantly reduced in *Its1n1* mutant synapses but only at mature synapses (Figure 4). They demonstrate a rescue of this effect after injecting the DH-PH domain of intersectin-1 and suggest a mechanism whereby *Its1n1* becomes enriched near presynaptic voltage-gated calcium channels in mature

synapses in order to help replenish the RRP.

Overall, this is a well written manuscript on an important topic in the field of synaptic physiology, as mechanisms of RRP replenishment are not as well established as other aspects of synaptic transmission. Intersectin is a scaffolding protein with many functions at synapses and many interaction partners, and so it has been challenging to study. That the *Itsn1* mutants have highly specific effects on RRP replenishment, but not other aspects of neurotransmission, is interesting and provides compelling evidence for this newly-identified function of this multifunctional protein. The physiology data presented are very solid and include appropriate controls. However, the conclusions about the molecular mechanisms based on the imaging data are currently a bit overreaching in the absence of additional information. Specific points that need to be addressed are listed below:

1. Per the journal requirements, please add the Ethics Statement to the Methods and include all the points in the Animal Ethics checklist.
2. The Methods and Results need to include more information on the DH-PH experiments, including the amino acid peptide sequence, concentration, and what assurances that the DH-PH domain is functionally specific for *Itsn1* and not other DH-PH domain containing proteins. DH-PH domains are fairly well conserved and found in numerous proteins.
3. Similarly, the *Itsn1* mutant mice need to be described better in the Methods and Results, including how the mice were made and any prior characterization of synapses, including the calyx of Held synapses. The Western blot implies that this mutant is a null, and therefore loss of function. However, curiously, the immunofluorescence signal in Figure 1E is measurable in the *Itsn1* mutant mice (is this background?). This discrepancy needs to be clarified, i.e. whether the mutant is making any *Itsn1* protein, a protein fragment, or none at all.
4. To make the claim that the enriched proximity of VGCCs and intersectin at mature synapses is the "structural basis for age-dependent regulation of *Itsn1* in RRP replenishment", it is necessary to determine exactly where the synaptic vesicles and/or release sites are in relation to VGCCs and *Itsn1*. This could be accomplished with 3-color labeling using the vGlut1 antibody or other synaptic vesicle or active zone markers. This statement makes the assumption that the calcium channels are not being redistributed relative to the release sites upon synaptic maturation, but the citrine-*Cacna1a* imaging does show altered, more punctate localization at P16-17.
5. Related to the point above, how can the authors rule out the alternative hypothesis that the reduced RRP replenishment in *Itsn1* mutants is somehow due to the abolished increase in vGlut1 that is normally observed at WT synapses, that is, a more general issue with synapse development? This is raised at the end of the Discussion but was not adequately discussed nor addressed either with experiments.

END OF COMMENTS

Confidential Review

03-May-2024

Response to Editor and Referee Comments on Manuscript JP-RP-2024-286462

Overall response: We wish to thank the editor and reviewers for their invaluable feedback on our manuscript. We are encouraged by their recognition of our work as “highly impactful”, with “strong and convincing” data supporting an age-dependent role of intersectin-1 (Itsn1) in replenishment of the readily releasable pool (RRP) of synaptic vesicles (SVs). To address the critiques, we have revised the manuscript with all changes highlighted in blue font. Below are our explanations for the revisions. The editor’s and referees’ comments are italicized, followed by our response to each point.

Reviewing Editor:

This manuscript is focused on the molecular mechanisms that regulate RRP replenishment. In this manuscript the authors demonstrate the Intersectin 1 is a key regulatory protein in RRP replenishment. Both reviewers found the data important, rigorous, and the findings highly impactful. Both reviewers had specific comments with respect to IHC analysis and correlation to western blot findings. It was pointed out that the western blots show no Intersectin protein, but IHC shows ~50% reduction in protein levels at the calyx. The authors need to address this discrepancy. Furthermore, both commented that conclusions and interpretations of results are a bit overreaching, and conclusions/interpretations should be toned down. In addition, Reviewer#1 has a specific comment with respect to the title. These comments can be addressed by carefully revising and rewriting their manuscript. In addition, please include an Ethics statement in the methods sections as required by Journal of Physiology policy.

Please also see “Required items” above.

Author Response: We sincerely appreciate your support of our manuscript. To address the discrepancy between Western blot (WB) and immunohistochemistry (IHC) results, we clarified that Itsn1 mutation caused a complete loss of Itsn1 expression, evidenced by the absence of Itsn1 bands in WB (Fig. 1A) and Itsn1 signals in IHC images (Fig. 1C). In our previous submission, we did not subtract the background using offset control during image quantification, which led to high fluorescence intensity in Itsn1 mutant synapses due to autofluorescence (old Fig. 1E). We have corrected this in the new Fig. 1E, showing significantly reduced Itsn1 signals in the mutant groups. Also, we’d like to emphasize that WB and IHC differ fundamentally in protein detection: WB detects 2D structures in denatured samples, while IHC detects 3D structures in intact tissues. Due to differences in antibody affinities for 2D vs 3D targets, secondary antibodies, buffer solutions, and other experimental conditions, WB and IHC results are not directly comparable but should be assessed collectively. The consistent reduction of Itsn1 detected by both methods confirms the Itsn1 deletion.

Furthermore, we have toned down the conclusions to avoid overreaching and changed the title as suggested by Referee #1. We also included Ethical Approval in the Methods and provided other items required by the journal policy.

Referee #1:

This paper determines how Itsn1 regulates synaptic transmission during development. To answer the question, the authors investigated transmitter release at the calyx of Held synapse in young and old mice, both wild-type and Itsn1-deficient mutant mice. Remarkably, the physiological consequences of loss of Itsn1 are quite specific: while virtually every measured presynaptic and postsynaptic property is identical in WT and mutant mice, there is a selective slowing of the fast, Ca²⁺-dependent component of recovery from synaptic depression. Even more remarkable is that this phenotype is observed only in older mice, at an age where the calyx synapse is mature. This developmental switch in the role of Itsn1 is correlated with an apparent change in the spatial relationship between Itsn1 and P/Q-type calcium channels. The authors conclude that the colocalization of Itsn1 and calcium channels is critical for the replenishment of the readily-releasable pool of synaptic vesicles following depletion of this pool during synaptic depression.

The physiological data in the paper are strong and convincing; I am impressed by the specificity of the physiological phenotype. The histological data are OK, as far as they go, and in any case are not the main point of the paper (but see point 1 below).

Author Response: We sincerely appreciate your enthusiasm and your insightful comments about our study. In response to your critiques, we had changed the title, added GAPDH as a control in Western blots (WB), explained the discrepancy between WB and immunohistochemistry (IHC) results, elaborated on the effect of the DH-PH peptide, and expanded the Discussion section.

Here are some suggestions for improvement that the authors should consider:

1. Title: "Developmental translocation of Intersectin-1 pivots calcium-dependent replenishment of the readily releasable pool of synaptic vesicles". I am not sure what "pivots" means in the context of the data shown in the paper. A clearer/more literal title would be advisable, because the authors have not really established any causal relationship between the observed developmental changes in Itsn1 localization and the Itsn1-related changes in recovery from synaptic depression. For example, one could argue that the changes in Itsn1 localization reflect changes in the location of sites of endocytosis and are unrelated to the location of calcium channels (or sites of exocytosis or rapid RRP replenishment).

Author Response: Thank you for your suggestion. The title is changed to “Intersectin-1 enhances calcium-dependent replenishment of the readily releasable pool of synaptic vesicles over development”.

2. In Figure 1A-B, including an internal control for the expression level of Itsn1 would be advantageous to control for possible differences in loading. Reference proteins, such as GAPDH, tubulin, or actin, are typically used. At a minimum, the procedure used for normalization in Fig. 1B should be explained more thoroughly.

Author Response: As suggested, we have added GAPDH as a loading control in the WB experiments and re-analyzed the data by normalizing Itsn1 and dynamin signals to GAPDH (Fig. 1 & 6). Additionally, we have ensured that each sample contained the same amount of protein (3.2 µg), as detailed in the Methods section (Page 11).

3. P. 13, para. 3: "brain tissue containing the medial nucleus of the trapezoid body" - roughly what fraction of the sample does the MNTB represent? I am guessing that it is a pretty minor part of the total sample volume; if so, authors might just want to say, "brainstem tissue".

Author Response: We apologize for the confusion. The MNTB region used for WB analysis was micro-dissected with fine needles under a microscope. Thus, the brain tissue primarily contained MNTB nuclei. This has been clarified in the Methods (Page 10) and the Results (Page 15).

4. Is there any reason to think that Itsn2 could compensate from the loss of Itsn1, particularly in the immature calyx?

Author Response: Thank you for the interesting point. As there is no evidence for functional redundancy between Itsn1 and Itsn2, it is unlikely that Itsn2 can compensate for the loss of Itsn1 at any developmental stage. We have discussed this as follows “Although Itsn1 and Itsn2 are homologs with overlapping tissue expression (Gubar *et al.*, 2013; Herrero-Garcia & O'Bryan, 2017), there is no evidence of functional redundancy between them. In mouse models, Itsn1 deletion leads to learning impairments (Sengar *et al.*, 2013), a reduced midline corpus callosum (Sengar *et al.*, 2013), and synaptic dysfunctions (Sakaba *et al.*, 2013; Gerth *et al.*, 2017; Japel *et al.*, 2020). In contrast, Itsn2-knockout mice do not show these deficits (Sengar *et al.*, 2013; Gerth *et al.*, 2017). The disparity may be due to the enrichment of the Itsn1 long isoform in neurons (Hussain *et al.*, 1999). Thus, the developmental effect of Itsn1 on RRP replenishment (Fig. 4) is unlikely caused by a compensatory action of Itsn2, though future studies are required to confirm this.” (Page 26-27).

5. P. 14, para. 1: Do we know what Itsn1 is doing in the soma of postsynaptic MNTB neurons? The signal is pretty strong, so it must be doing something.

Author Response: Thank you for another interesting point. Although not tested in this study, we have discussed the potential role of Itsn1 in postsynaptic MNTB neurons as follows “Itsn1 is present in both pre- and postsynaptic compartments of the calyx of Held synapse throughout development (Fig. 1 and 8). However, basal synaptic transmission, including spontaneous and synchronized transmitter release, remains unchanged in Itsn1-lacking synapses (Fig. 2 and 3), suggesting that Itsn1 does not affect postsynaptic glutamate receptors mediating these events. In hippocampal neurons, Itsn1 is known to regulate dendritic spine development (Nishimura *et al.*, 2006; Thomas *et al.*, 2009) and long-term plasticity (Jakob *et al.*, 2017), likely by interacting with cell signaling pathways involved in neuronal migration, differentiation, and survival

(Mintoo *et al.*, 2024). It remains to be determined if Itsn1 plays a role in activity-dependent plasticity at postsynaptic MNTB neurons (Joshi *et al.*, 2007).” (Page 27).

6. P. 14, line 8: *I see no need to list all of the numerical data that are associated with Fig. 1E (and other figures). The whole point of a figure is to visually represent the data, so there is no need for repetition unless some precise numerical comparison is intended. Such lists of extraneous numbers here (and throughout the paper) significantly interrupt the flow for readers. Also, if the authors really want to leave in the numbers there is no need for 4 significant figures - there is no biological measurement that has an accuracy of 0.01%.*

Author Response: Thank you for the suggestion. We have removed all numbers throughout the manuscript, except for Fig. 4-6, where we analyze the recovery curves. The parameters such as $P_{\text{fast}}/P_{\text{slow}}$ and $\tau_{\text{fast}}/\tau_{\text{slow}}$ in the Results represent the average data from fitting individual cells with exponential cumulative functions, while the parameters in the figures are derived from fitting the average curves. Because of this difference, we have kept the numerical data. Wherever numbers are included, they contain only 3 significant figures in compliance with the journal policy.

7. Fig. 1E: *I do not understand why the fluorescence signals for Itsn1 are only reduced by 50% in the mutants? Both the Western blots shown in Fig. 1A, and the fluorescence images shown in Fig. 1D suggest that these signals should be vanishingly small in the mutants.*

Author Response: We apologize for this error. In our previous submission, we did not subtract the background using offset control during image quantification, which resulted in artificially high fluorescence intensity in Itsn1 mutant synapses due to autofluorescence (old Fig. 1E). We have corrected this in the new Fig. 1E, showing significantly reduced Itsn1 signals in the mutant groups. The Methods (Page 13) and Results (Page 15) sections are updated accordingly.

8. P. 17, para. 2: *Fig. 5A is not really explained in the text.*

Author Response: Thank you for pointing this out. Fig. 5A is now described in the Results section (Page 18).

9. P. 18, para. 2 and Figures 6A and 6B: *My only serious scientific concern, in the entire paper, is interpretation of the effects of the DH-PH peptide. Specifically, is this peptide exerting a dominant-negative or gain-of-function effect? The authors clearly interpret their data in terms of the latter possibility, but in fact most small peptides serve as inhibitors of protein-protein interactions, rather than activators. The Hussein et al. (2001) paper used full-length DH-PH (~350 amino acids) and saw that this activated Cdc42. Did the authors use the same construct (a fragment or domain, rather than a peptide) or a smaller synthetic peptide (~20-30 amino acids)?*

I could not find any explicit description of the "peptide" that the authors used or how they synthesized or expressed it. Also, if Itsn1 works as a Ca²⁺ sensor, as suggested in the Discussion, how would DH-PH rescue in the mutant given that this fragment/peptide lacks the C2 domain of Itsn1?

Author Response: Thank you for raising this important point. We have provided information on the peptide synthesis and sequence in the Methods (Page 10) and the peptide application in the Results and figure legend. The peptide contained the full-length DH-PH domains, as used in the Hussain et al. (2001) study. In our case, the DH-PH peptide likely had a gain-of-function effect, facilitating RRP replenishment in Itsn1-lacking synapses. We have also discussed the specificity of this peptide on Page 26.

Regarding the roles of Itsn1 domains in RRP replenishment, we believe multiple domains could be involved and are not mutually exclusive. For example, the C2 domain may sense local Ca²⁺ changes and initiate GEF activity of the DH-PH domains, thereby regulating Cdc42 in actin polymerization, which is critical for RRP replenishment (Sakaba et al., 2013). However, future experiments are needed to elucidate the domain interactions. We have discussed the possibilities within the scope of our study (Page 25-26).

10. Fig. 8D: I am wondering about the resolution of the confocal microscope used for these imaging experiments. The authors state that the confocal microscope used has ~200 nm resolution (p. 13, top), which is consistent with the Abbe diffraction limit (245 nm for 520 nm emission collected with a 1.4 NA objective). However, the Cacna1a (Citrine) fluorescence shown in Fig. 8D appears to occupy only a single bin (width 100 nm) and it seems unlikely that the entire distribution of this signal is less than half of the theoretical minimum. Also, note that the resolution should be poorer (approximately 300 nm) for Itsn1 imaging, because of the longer wavelength of the fluorophore used (A647). Thus, it is possible that the first bin for the "presynaptic" Itsn1 signal (to the right of the blanked-out bin in the bottom graph of Fig. 8D) still could be coming from the postsynaptic side, which is less than 300 nm away. The bottom line is that the authors should exercise some caution in interpreting the data shown in Fig. 8D.

Author Response: Thank you for the thoughtful comment. Since the theoretical optical resolution of the Leica TCS SP8 confocal imaging system using a 1.4 NA objective is 184 and 231 nm at 515 and 648 nm excitations, respectively, we detected pre- and postsynaptic Itsn1 maxima with a 200 nm separation (Fig. 8D). Each line plot in every detection channel (515 / 598 / 648 nm) was aligned with the maxima of Cacna1a^{Citrine} signals (at 0 μm). We acknowledge that the presynaptic A647 fluorescence signal is still partly contaminated by postsynaptic signal and vice versa, even after excluding a 200-nm region around the Cacna1a^{Citrine} maxima. Yet, the distinct distribution of immature and mature Itsn1 maxima in the presynaptic compartment, but not in the postsynaptic compartment, indicates that the postsynaptic signal contribution to the presynaptic difference is minimal. Additionally, we have corrected an error in plotting the Itsn1 histogram, which was incorrectly plotted to the beginning of the bin range, visually narrowing

the separation between pre- and postsynaptic sides. We have now re-plotted the bins to the bin center (Fig. 8D). Furthermore, we selected VGCC clusters at the inner edge of A594d-labeled calyces (typically at the largest calyx cross-section) to enhance the optical separation of pre- and postsynaptic sides in the x-y dimension. Based on these considerations, we have revised the Methods (Page 14), Results (Page 22) and Figure legend sections (Page 35).

Finally, here are some very minor suggestions for improving word usage:

p. 3, last line: "it converts excitatory inputs from globular bushy cells on one side to inhibitory outputs to the superior olive on the other side." Perhaps change "on one side" to "ipsilateral" and "other side" to "contralateral".

Author Response: Thank you for your suggestion. This sentence is changed to “It converts excitatory inputs from ipsilateral globular bushy cells into inhibitory outputs to the contralateral superior olive,” (Page 3).

p. 4, middle of the page: "alternation" probably should be "alteration".

Author Response: This is corrected. Thank you!

p. 5, line 7: "clathrin mediated" should be hyphenated.

Author Response: This is corrected. Thank you!

p. 14, lines 1-3: "To assess Itsn1 location, we co-labeled Itsn1 with a presynaptic marker vesicular glutamate transporter-1". Itsn1 was not co-labelled; a second independent label, vGlut1, was used to examine the location of synaptic vesicles.

Author Response: We apologize for the confusing sentence. It is now corrected: “To examine Itsn1 location in the MNTB, we stained brainstem slices with antibodies specific for Itsn1 and a presynaptic marker vGlut1.” (Page 15).

p. 14, para. 2: "Given that Itsn1 mutation correlated with reduced vGlut1 at the calyx of Held synapse" this correlation was only observed in older mice, so some rewording seems necessary.

Author Response: This is reworded as such “Given the reduced vGlut1 expression in mature Itsn1-Mut synapses” (Page 15). Thank you!

p. 15, para. 2: "As each postsynaptic neuron is typically innervated by one calyx, these EPSCs were large and occurred in an all-or-none manner". Probably should remove (or relocate) the "large" part, because the size of the EPSC is unrelated to innervation by one calyx.

Author Response: The irrelevant statement is removed. Thank you!

Referee #2:

Synaptic transmission requires reliable exocytosis and endocytosis across a range of stimulation frequencies. At high frequencies of activity, the readily-releasable pool (RRP) of synaptic vesicles becomes depleted and must be rapidly replenished to maintain neurotransmission. This study by Yang et al., documents an age-dependent role for intersectin-1 (Itsn1) in calcium-dependent recovery from short term depression at the calyx of Held synapse in the auditory brainstem. The authors delineate a number of aspects of synaptic transmission that are unchanged in the Itsn1 mutants, including calcium entry, action potentials, spontaneous and evoked EPSCs, and RRP size. However, a key finding is that the RRP replenishment after calcium-dependent short term depression is significantly reduced in Itsn1 mutant synapses but only at mature synapses (Figure 4). They demonstrate a rescue of this effect after injecting the DH-PH domain of intersectin-1 and suggest a mechanism whereby Itsn1 becomes enriched near presynaptic voltage-gated calcium channels in mature synapses in order to help replenish the RRP.

Overall, this is a well written manuscript on an important topic in the field of synaptic physiology, as mechanisms of RRP replenishment are not as well established as other aspects of synaptic transmission. Intersectin is a scaffolding protein with many functions at synapses and many interaction partners, and so it has been challenging to study. That the Itsn1 mutants have highly specific effects on RRP replenishment, but not other aspects of neurotransmission, is interesting and provides compelling evidence for this newly-identified function of this multifunctional protein. The physiology data presented are very solid and include appropriate controls. However, the conclusions about the molecular mechanisms based on the imaging data are currently a bit overreaching in the absence of additional information. Specific points that need to be addressed are listed below:

Author Response: We sincerely appreciate your enthusiasm and your insightful comments about our study. In response to your critiques, we have added the Ethical Approval, elaborated on the DH-PH peptide and Itsn1 mutant mice, and discussed potential mechanisms underlying Itsn1-mediated replenishment of the RRP.

1. Per the journal requirements, please add the Ethics Statement to the Methods and include all the points in the Animal Ethics checklist.

Author Response: Thank you for the suggestion. We have included Ethical Approval and other details related to lab animal use, according to the journal policy (Page 6).

2. The Methods and Results need to include more information on the DH-PH experiments, including the amino acid peptide sequence, concentration, and what assurances that the DH-PH domain is functionally specific for Itsn1 and no other DH-PH domain containing proteins. DH-PH domains are fairly well conserved and found in numerous proteins.

Author Response: Thank you for raising this important point. The peptide synthesis, sequence, and concentration are now included in the Methods (Page 10) and Results (Page 19) sections. We also discussed the specificity of the DH-PH peptide (Page 26). Briefly, the domain specificity for each protein is determined by sequence variation, 3D structure, and binding partners/substrates in a particular intracellular environment. The ability of the peptide containing the full-length DH-PH domains to rescue the exact phenotype resulting from Itsn1 deletion supports its specific function, likely substituting Itsn1 protein. But future studies are required to confirm this.

3. Similarly, the Itsn1 mutant mice need to be described better in the Methods and Results, including how the mice were made and any prior characterization of synapses, including the calyx of Held synapses. The Western blot implies that this mutant is null, and therefore loss of function. However, curiously, the immunofluorescence signal in Figure 1E is measurable in the Itsn1 mutant mice (is this background?). This discrepancy needs to be clarified, i.e. whether the mutant is making any Itsn1 protein, a protein fragment, or none at all.

Author Response: Thank you for raising another important point. We have included details on the generation of Itsn1 mutant mice in the Methods (Page 6) and Results (Page 15). These mice were created using a gene-trap embryonic stem cell line, leading to a complete loss of function of the *Itsn1* gene and the elimination of both short and long isoforms in the whole brain (Sengar et al., 2013). For the first time, we demonstrated Itsn1 loss at the calyx of Held synapse using Western blot (Fig. 1A) and immunohistochemistry (IHC) analyses (Fig. 1C).

In our previous submission, we did not subtract the background using offset control during image quantification, which resulted in artificially high fluorescence intensity in Itsn1 mutant synapses due to autofluorescence (old Fig. 1E). We have corrected this in the new Fig. 1E, which now shows significantly reduced Itsn1 signals in the mutant groups. The Methods (Page 13) and Results (Page 15) sections are updated accordingly.

4. To make the claim that the enriched proximity of VGCCs and Intersectin at mature synapses is the "structural basis for age-dependent regulation of Itsn1 in RRP replenishment", it is necessary to determine exactly where the synaptic vesicles and/or release sites are in relation to VGCCs and Itsn1. This could be accomplished with 3-color labeling using the vGlut1 antibody or other synaptic vesicle or active zone markers. This statement makes the assumption that the calcium

channels are not being redistributed relative to the release sites upon synaptic maturation, but the citrine-Cacna1a imaging does show altered, more punctate localization at P16-17.

Author Response: We agree with the Reviewer that voltage-gated Ca²⁺ channels (VGCCs) redistribute relative to the release site during development, as the coupling distance between VGCCs and readily-releasable synaptic vesicles shortens from 30 nm at postnatal day (P) 7 to 20 nm at P14 (Nakamura et al., 2015). However, high-resolution electron microscopic experiments (SDS-FRL) using RIM and Cav2.1 immuno-gold particles have shown that VGCCs are localized within the active zone (Nakamura et al., 2015, Figure S2), making release sites and VGCCs indistinguishable under a confocal microscope. So, we used VGCC clusters as an indicator of release sites. Acknowledging this technical limitation, we have toned down our conclusions and changed “structural basis for” to “may underlie” the age-dependent regulation of Itsn1 in RRP replenishment (Pages 2 and 22).

5. Related to the point above, how can the authors rule out the alternative hypothesis that the reduced RRP replenishment in Itsn1 mutants is somehow due to the abolished increase in vGlut1 that is normally observed at WT synapses, that is, a more general issue with synapse development? This is raised at the end of the Discussion but was not adequately discussed nor addressed either with experiments.

Author Response: Thank you for pointing this out. We have elaborated on the potential effect of reduced vGlut1 in mature Itsn1 mutant synapses in the Discussion (Page 27-28), which reads “Despite the fact that vGlut1 enhances SV refilling with glutamate (Nakakubo *et al.*, 2020), its reduced level at Itsn1-Mut synapses might not directly contribute to the slower RRP replenishment, as vGlut1-knockout synapses exhibit faster RRP replenishment (Nakakubo *et al.*, 2020). This indicates that the RRP replenishment is a different process from loading glutamate into SVs, although developmental modulation of the synapse by vGlut1 cannot be excluded.”

Dear Dr Yang,

Re: JP-RP-2024-286462R1 "Intersectin-1 enhances calcium-dependent replenishment of the readily releasable pool of synaptic vesicles over development" by Yi-Mei Yang, Adam Fekete, Jason Arsenault, Ameet S Sengar, Jamila Aitoubah, Giovanbattista Grande, Angela Li, Eric W Salter, Alex Wang, Melanie D Mark, Stefan Herlitze, Sean E Egan, Michael W. Salter, and Lu-Yang Wang

Thank you for submitting your manuscript to The Journal of Physiology. It has been assessed by a Reviewing Editor and by 2 expert referees and we are pleased to tell you that it is acceptable for publication following satisfactory revision.

REVISION CHECKLIST:

Please upload two versions of your manuscript text: one with all relevant changes highlighted and one clean version with no changes tracked. The manuscript file should include all tables and figure legends, but each figure/graph should be uploaded as separate, high-resolution files. The journal is now integrated with Wiley's Image Checking service. For further details,

see: <https://www.wiley.com/en-us/network/publishing/research-publishing/trending-stories/upholding-image-integrity-wileys-image-screening-service>

We look forward to receiving your revised submission.

Yours sincerely,

Katalin Toth
Senior Editor
The Journal of Physiology

REQUIRED ITEMS

- You must start the Methods section with a paragraph headed Ethical Approval. A detailed explanation of journal policy and regulations on animal experimentation is given in Principles and standards for reporting animal experiments in The Journal of Physiology and Experimental Physiology by David Grundy J Physiol, 593: 2547-2549. doi:10.1113/JP270818). A checklist outlining these requirements and detailing the information that must be provided in the paper can be found at: <https://physoc.onlinelibrary.wiley.com/hub/animal-experiments>. Authors should confirm in their Methods section that their experiments were carried out according to the guidelines laid down by their institution's animal welfare committee, and conform to the principles and regulations as described in the Editorial by Grundy (2015), including an ethics approval reference number. The Methods section must contain a statement about access to food, water and housing, details of the anaesthetic regime: anaesthetic used, dose and route of administration, and method of killing the experimental animals.

EDITOR COMMENTS

Reviewing Editor:

The authors have done an excellent job of responding to the reviewers comments. Please make the nomenclature change as suggested by reviewer#1.

To comply with our ethics policy, please include the method of euthanasia in the Methods section.

Senior Editor:

Please respond to the final comments of the Reviewing Editor and Referee 1.

REFEREE COMMENTS

Referee #1:

The revisions have made this good paper even better.

My only remaining scientific concern is nomenclature: confusion results from calling the 350-mer DH-PH domain a "peptide". While it is true that even a full-length protein is technically a "peptide", typically the term "peptide" is reserved for small polypeptides (e.g. 20 or so amino acids). I recommend calling this a "domain" or "fragment", rather than a "peptide".

Referee #2:

The authors have done a commendable job addressing the prior critiques. This study strengthens our understanding of the function of intersectin-1 at synapses during development. Moreover, by elucidating the mechanisms that contribute to Ca²⁺-dependent recovery from short-term depression, this work provides a unique and important contribution to the field of synaptic transmission. The data are impeccable and the manuscript well written. I have no further comments to share at this time.

END OF COMMENTS

1st Confidential Review

13-Aug-2024

Response to Editor and Referee Comments on Manuscript JP-RP-2024-286462R1

Overall response: We thank the editors and reviewers for their valuable feedback, which has helped us to improve the manuscript. To address their comments, we have revised the text, with all changes highlighted in blue. Below are our explanations of the revisions. The editors' and reviewers' comments are italicized, followed by our response to each point.

EDITOR COMMENTS

Reviewing Editor:

The authors have done an excellent job of responding to the reviewers comments. Please make the nomenclature change as suggested by reviewer#1.

Author Response: We sincerely appreciate your enthusiasm and insightful comments on our study. They have greatly helped us to improve the quality of our manuscript. In response to the Reviewer #1's suggestion, we have replaced the term "peptide" with "domain" or "fragment" where appropriate.

To comply with our ethics policy, please include the method of euthanasia in the Methods section.

Author Response: Thank you for raising this issue. We have now added a description of the euthanasia procedure in the Methods section (Page 7), which states "Decapitation was used to collect brain tissue. The mouse was restrained in a plastic cone and quickly decapitated with a sharp single-use blade (009 RD; VWR). This method ensured rapid loss of consciousness and protected the tissue from chemical contamination".

Senior Editor:

Please respond to the final comments of the Reviewing Editor and Referee 1.

Author Response: We sincerely appreciate your support of our manuscript. In response to the Reviewing Editor's comment, we have added a description of the euthanasia procedure in the Methods section (Page 7), which states "Decapitation was used to collect brain tissue. The mouse was restrained in a plastic cone and quickly decapitated with a sharp single-use blade (009 RD; VWR). This method ensured rapid loss of consciousness and protected the tissue from chemical contamination".

In response to the Referee #1's comment, we have replaced the term "peptide" with "domain" or "fragment" where appropriate.

REFEREE COMMENTS

Referee #1:

The revisions have made this good paper even better. My only remaining scientific concern is nomenclature: confusion results from calling the 350-mer DH-PH domain a "peptide". While it is true that even a full-length protein is technically a "peptide", typically the term "peptide" is reserved for small polypeptides (e.g. 20 or so amino acids). I recommend calling this a "domain" or "fragment", rather than a "peptide".

Author Response: We sincerely appreciate your enthusiasm and insightful comments on our study. They have greatly helped us to improve the quality of our manuscript. Following your suggestion, we have replaced the term “peptide” with “domain” or “fragment” where appropriate (blue font).

Referee #2:

The authors have done a commendable job addressing the prior critiques. This study strengthens our understanding of the function of intersectin-1 at synapses during development. Moreover, by elucidating the mechanisms that contribute to Ca²⁺-dependent recovery from short-term depression, this work provides a unique and important contribution to the field of synaptic transmission. The data are impeccable and the manuscript well written. I have no further comments to share at this time.

Author Response: We sincerely appreciate your enthusiasm and insightful comments on our study. They have greatly helped us to improve the quality of our manuscript.

Dear Dr Yang,

Re: JP-RP-2024-286462R2 "Intersectin-1 enhances calcium-dependent replenishment of the readily releasable pool of synaptic vesicles over development" by Yi-Mei Yang, Adam Fekete, Jason Arsenault, Ameet S Sengar, Jamila Aitoubah, Giovanbattista Grande, Angela Li, Eric W Salter, Alex Wang, Melanie D Mark, Stefan Herlitze, Sean E Egan, Michael W. Salter, and Lu-Yang Wang

We are pleased to tell you that your paper has been accepted for publication in The Journal of Physiology.

Authors should note that it is too late at this point to offer corrections prior to proofing. Major corrections at proof stage, such as changes to figures, will be referred to the Editors for approval before they can be incorporated. Only minor changes, such as to style and consistency, should be made at proof stage. Changes that need to be made after proof stage will usually require a formal correction notice.

If you would like to receive our 'Research Roundup', a monthly newsletter highlighting the cutting-edge research published in The Physiological Society's family of journals (The Journal of Physiology, Experimental Physiology and Physiological Reports), please click this link, fill in your name and email address and select 'Research Roundup': <https://www.physoc.org/journals-and-media/membernews/>.

Yours sincerely,

Katalin Toth
Senior Editor
The Journal of Physiology

P.S. - You can help your research get the attention it deserves! Check out Wiley's free Promotion Guide for best-practice recommendations for promoting your work at www.wileyauthors.com/eeo/guide. You can learn more about Wiley Editing Services which offers professional video, design, and writing services to create shareable video abstracts, infographics, conference posters, lay summaries, and research news stories for your research at www.wileyauthors.com/eeo/promotion.

IMPORTANT NOTICE ABOUT OPEN ACCESS: To assist authors whose funding agencies mandate public access to published research findings sooner than 12 months after publication, The Journal of Physiology allows authors to pay an Open Access (OA) fee to have their papers made freely available immediately on publication.

You can check if your funder or institution has a Wiley Open Access Account here: <https://authorservices.wiley.com/author-resources/Journal-Authors/licensing-and-open-access/open-access/author-compliance-tool.html>.